# Shallow Flow Matching for Coarse-to-Fine Text-to-Speech Synthesis

**Dong Yang[1]\*, Yiyi Cai[2], Yuki Saito[1], Lixu Wang[3], Hiroshi Saruwatari[1]**
[1]The University of Tokyo, [2]Independent Researcher, [3]Nanyang Technological University
\*ydqmkkx@gmail.com

## Abstract

We propose Shallow Flow Matching (SFM), a novel mechanism that enhances flow matching (FM)-based text-to-speech (TTS) models within a coarse-to-fine generation paradigm. Unlike conventional FM modules, which use the coarse representations from the weak generator as conditions, SFM constructs intermediate states along the FM paths from these representations. During training, we introduce an orthogonal projection method to adaptively determine the temporal position of these states, and apply a principled construction strategy based on a single-segment piecewise flow. The SFM inference starts from the intermediate state rather than pure noise, thereby focusing computation on the latter stages of the FM paths. We integrate SFM into multiple TTS models with a lightweight SFM head. Experiments demonstrate that SFM yields consistent gains in speech naturalness across both objective and subjective evaluations, and significantly accelerates inference when using adaptive-step ODE solvers. Demo and codes are available at `https://ydqmkkx.github.io/SFMDemo/`.

## 1 Introduction

Text-to-speech (TTS) synthesis has advanced with generative algorithms in recent years, particularly autoregressive (AR) [45, 25, 22, 49, 38, 46], diffusion [33, 12, 48, 2, 24], and flow matching (FM) [23, 13, 15, 10, 30, 7, 42] methods. TTS models typically contain a front-end for processing contextual information and a back-end for speech synthesis. Due to the strong temporal alignment between texts and speech, many diffusion- or FM-based TTS models adopt a coarse-to-fine generation paradigm: a weak generator first produces coarse representations conditioned on the input context, which are then refined by the diffusion or FM module into high-quality mel-spectrograms, and finally converted into audio waveforms by a vocoder. There are two main approaches to the weak generator. One [33, 13, 15, 30] follows traditional TTS designs, where a non-autoregressive encoder and an alignment module [35, 19] jointly generate coarse mel-spectrograms when aligning encoder outputs with target mel-spectrograms. The other [2, 8, 9, 31, 51] employs an AR large language model (LLM) as a context processor and weak generator to generate discrete speech tokens, which are transformed into continuous mel-spectrograms by a diffusion or FM module.

FM has gained increasing attention due to its efficiency and high-quality generation in TTS. In conventional coarse-to-fine FM-based TTS models, coarse representations are used as conditions for the flow module. However, the generation still starts from pure noise, resulting in a suboptimal allocation of modeling capacity. Since the coarse representations already encode a significant portion of the overall semantic and acoustic structure, modeling the early stage from noise becomes redundant and contributes little to the quality of the final output. To deal with this issue, DiffSinger [27] has introduced a shallow diffusion mechanism on singing voice synthesis (SVS) and TTS. It uses a simple mel-spectrogram decoder, with the diffusion module starting generation at a shallow step based on the output of the decoder. Therefore, we extend the idea of shallow diffusion mechanism (with the name)

on FM and propose shallow flow matching (SFM) for coarse-to-fine TTS models. During training, we construct intermediate states along the FM paths based on the coarse representations. To achieve this, we employ an orthogonal projection method to adaptively determine the corresponding time, define a principled construction approach, and formulate a single-segment piecewise flow. During inference, the generation starts from the intermediate state, skipping the early stages and focusing computation on the latter part of the flow. This leads to more stable generation and enhanced synthesis quality, and accelerates inference when using adaptive-step ordinary differential equation (ODE) solvers.

We validate the proposed SFM method on multiple coarse-to-fine TTS models, covering two mainstream FM architectures, U-Net [36] and DiT [32]. Experimental results show that SFM consistently improves the naturalness of synthesized speech, as measured by both pseudo-MOS metrics and subjective evaluations. We also observe significant inference acceleration across various adaptive-step ODE solvers.

## 2 Preliminaries

### 2.1 Flow matching

A time-dependent diffeomorphic map $\phi_t : [0, 1] \times \mathbb{R}^d \to \mathbb{R}^d$ describes the smooth and invertible transformation of data points $\boldsymbol{x} \in \mathbb{R}^d$ over time $t \in [0, 1]$, then a *flow* is defined via the ODE with a time-dependent *vector field* (VF) $\boldsymbol{u}_t : [0, 1] \times \mathbb{R}^d \to \mathbb{R}^d$:

$$\boldsymbol{x}_t = \phi_t(\boldsymbol{x}_0), \quad \frac{d}{dt}\phi_t(\boldsymbol{x}_0) = \boldsymbol{u}_t(\phi_t(\boldsymbol{x}_0)). \tag{1}$$

VF $\boldsymbol{u}_t$ induces a *probability density path* $p_t : [0, 1] \times \mathbb{R}^d \to \mathbb{R}_{>0}$, which is a time-dependent probability density function (PDF). From time 0 to time $t$, the PDF of $\boldsymbol{x}$ is transported from $p_0(\boldsymbol{x}_0)$ to $p_t(\boldsymbol{x}_t)$ along $\boldsymbol{u}_t$. Chen et al. [6] proposed continuous normalizing flow (CNF) that models the $\boldsymbol{u}_t$ by a neural network $\boldsymbol{v}_{\boldsymbol{\theta}}(\boldsymbol{x}_t, t)$, where $\boldsymbol{\theta}$ are learnable parameters. A CNF reshapes a simple prior distribution $p_0$ into a complicated distribution $p_1$. Lipman et al. [26] further proposed flow matching (FM), whose objective is $\mathcal{L}_{\text{FM}} = \mathbb{E}_{t,p_t(\boldsymbol{x}_t)}\|\boldsymbol{v}_{\boldsymbol{\theta}}(\boldsymbol{x}_t, t) - \boldsymbol{u}_t(\boldsymbol{x}_t)\|^2$. Since appropriate $p_t$ and $\boldsymbol{u}_t$ are unknown, [26] constructed probability paths conditioned on data sample $\boldsymbol{x}_1 \sim q(\boldsymbol{x}_1)$. Specifically, let $p_0(\boldsymbol{x}_0) = \mathcal{N}(\boldsymbol{x}_0|\boldsymbol{0}, \boldsymbol{I})$, $p_1(\boldsymbol{x}_1) \approx q(\boldsymbol{x}_1)$, the conditional probability path is $p_t(\boldsymbol{x}_t|\boldsymbol{x}_1) = \mathcal{N}(\boldsymbol{x}_t|\boldsymbol{\mu}_t(\boldsymbol{x}_1), \sigma_t(\boldsymbol{x}_1)^2 \boldsymbol{I})$. The flow and VF are considered with the forms:

$$\phi_t(\boldsymbol{x}_t) = \boldsymbol{\sigma}_t(\boldsymbol{x}_1)\boldsymbol{x}_t + \boldsymbol{\mu}_t(\boldsymbol{x}_1), \quad \boldsymbol{u}_t(\boldsymbol{x}_t|\boldsymbol{x}_1) = \frac{\boldsymbol{\sigma}'_t(\boldsymbol{x}_1)}{\boldsymbol{\sigma}_t(\boldsymbol{x}_1)}(\boldsymbol{x}_t - \boldsymbol{\mu}_t(\boldsymbol{x}_1)) + \boldsymbol{\mu}'_t(\boldsymbol{x}_1), \tag{2}$$

where $\boldsymbol{\mu}_t : [0, 1] \times \mathbb{R}^d \to \mathbb{R}^d$ is the time-dependent mean and $\boldsymbol{\sigma}_t : [0, 1] \times \mathbb{R}^d \to \mathbb{R}_{>0}$ is the time-dependent scalar standard deviation (std). $f'$ denotes the derivative with respect to time, $f' = \frac{d}{dt}f$. Corresponding to the optimal transport (OT) displacement interpolant [29], the mean and std are:

$$\boldsymbol{\mu}_t(\boldsymbol{x}_1) = t\boldsymbol{x}_1, \quad \sigma_t(\boldsymbol{x}_1) = 1 - (1 - \sigma_{\min})t, \tag{3}$$

where $\sigma_{\min}$ is a sufficiently small value. Substituting Eq. (3) into Eq. (2), the conditional flow and VF take the form:

$$\phi_t(\boldsymbol{x}_0) = (1 - t)\boldsymbol{x}_0 + t(\boldsymbol{x}_1 + \sigma_{\min}\boldsymbol{x}_0), \quad \boldsymbol{u}_t(\boldsymbol{x}_t|\boldsymbol{x}_1) = (\boldsymbol{x}_1 + \sigma_{\min}\boldsymbol{x}_0) - \boldsymbol{x}_0. \tag{4}$$

Then, we can minimize the conditional flow matching (CFM) loss during training, which is proven by [26] to be equivalent to minimizing the FM loss $\mathcal{L}_{\text{CFM}} = \mathbb{E}_{t,p_t(\boldsymbol{x}_t)}\|\boldsymbol{v}_{\boldsymbol{\theta}}(\boldsymbol{x}_t, t) - \boldsymbol{u}_t(\boldsymbol{x}_t|\boldsymbol{x}_1)\|^2$.

During inference, we use an ODE solver to solve the integral $\boldsymbol{x}_{\text{pred}} = \boldsymbol{x}_0 + \int_0^1 \boldsymbol{v}_{\boldsymbol{\theta}}(\boldsymbol{x}_t, t)\, dt$.

### 2.2 Classifier-free guidance

In FM-based generative models, a typical approach for controlling the generation process is to incorporate the input condition $\boldsymbol{c}$ during training and inference. To improve diversity and fidelity, classifier-free guidance (CFG) [16] can be employed. During training, the condition $\boldsymbol{c}$ is randomly dropped, enabling the model to learn from both conditional and unconditional contexts. During inference, a hyperparameter called the CFG strength $\beta > 0$ is introduced to control the trade-off:

$$\boldsymbol{v}_{\boldsymbol{\theta},\text{CFG}}(\boldsymbol{x}_t, t, \boldsymbol{c}) = \boldsymbol{v}_{\boldsymbol{\theta}}(\boldsymbol{x}_t, t, \boldsymbol{c}) + \beta(\boldsymbol{v}_{\boldsymbol{\theta}}(\boldsymbol{x}_t, t, \boldsymbol{c}) - (\boldsymbol{v}_{\boldsymbol{\theta}}(\boldsymbol{x}_t, t)). \tag{5}$$

Specifically, the FM module runs two forward passes at each time step, once with $\boldsymbol{c}$ and once without.

### 2.3 Flow matching-based TTS models

Our backbone configurations involve three fully open-source TTS models.

**Matcha-TTS:** Matcha-TTS [30] is a non-AR FM-based TTS model that employs a conventional encoder-decoder architecture. The Transformer-based [43] encoder takes phonemes and speaker IDs (for multi-speaker training) as input, producing hidden states and predicted phoneme-level durations. The U-Net-based FM decoder receives the encoder's outputs and speaker embeddings as FM conditions and generates mel-spectrograms.

Because the encoder adopts the monotonic alignment search (MAS)-based alignment [18, 19, 33] for phoneme-spectrogram alignment, the encoder outputs are optimized towards the ground-truth mel-spectrograms (via the *prior loss* in the paper), resulting in coarse mel-spectrograms.

**CosyVoice:** CosyVoice [8] is a large zero-shot TTS model that consists of four components: the text encoder, the speech tokenizer, the LLM, and the FM module. The speech tokenizer extracts discrete speech tokens from mel-spectrograms of waveforms, while the text encoder processes textual inputs and aligns text encodings with speech tokens. The LLM, a Transformer decoder-based model, takes speaker embeddings, text encodings, and prompt speech tokens as input and generates target speech tokens in an AR way. Subsequently, the FM module, conditioned on speaker embeddings, target speech tokens, and masked mel-spectrograms, generates the target mel-spectrograms.

The FM module adopts an encoder-decoder structure. The Conformer-based [14] encoder encodes the speech tokens generated by the LLM into hidden states and linearly projects them to the same dimension as mel-spectrograms. These hidden states are then upsampled by a length regulator for token-spectrogram alignment and fed into the U-Net-based decoder, along with other conditions, for FM training and inference. Therefore, the upsampled hidden states can be explicitly supervised to coarse mel-spectrograms.

In CosyVoice, the speech tokenizer, LLM, and flow modules are trained independently. Therefore, in our experiments, we only train the flow module using speech tokens generated by the officially pre-trained speech tokenizer, while employing the officially pre-trained LLM during inference.

**StableTTS:** StableTTS [1] is an open-source TTS model in which both the encoder and FM decoder leverage DiT blocks. Its encoder employs MAS-based alignment and outputs coarse mel-spectrograms.

To assess the effectiveness of our method under different input types (text sequences and speech tokens) within the DiT architecture, we further adapt StableTTS as the FM module of CosyVoice. The resulting backbone TTS model is referred to as **CosyVoice-DiT** throughout our work.

## 3 Method

### 3.1 Theorems

Proofs of our theorems are provided in Appendix A. Theorem 2 can be derived from PeRFlow [47] that divides CondOT paths into several windows and conducts piecewise reflow [28] in each window.

**Theorem 1.** *For any random variable* $\boldsymbol{x}_m \sim \mathcal{N}(t_m \boldsymbol{x}_1, \sigma_m^2 \boldsymbol{I})$*, where* $t_m \in [0, \infty)$ *and* $\sigma_m \in (0, \infty)$*, we define a transformation that maps* $\boldsymbol{x}_m$ *onto the conditional OT (CondOT) paths. The output distribution varies continuously with respect to* $t_m$ *and* $\sigma_m$ *under the Wasserstein-2 metric.*

$$\Delta = (1 - \sigma_{\min})t_m + \sigma_m, \tag{6}$$

$$\boldsymbol{x}_\tau = \begin{cases} \sqrt{(1 - (1 - \sigma_{\min})t_m)^2 - \sigma_m^2}\boldsymbol{x}_0 + \boldsymbol{x}_m, & \text{if } \Delta < 1, \\ \frac{1}{\Delta}\boldsymbol{x}_m, & \text{if } \Delta \geq 1, \end{cases} \tag{7}$$

*where* $\boldsymbol{x}_0 \sim \mathcal{N}(\boldsymbol{0}, \boldsymbol{I})$*, with corresponding* $\tau = \min(t_m, \frac{t_m}{\Delta})$ *on the path.*

**Theorem 2.** *For arbitrary intermediate states on the CondOT paths:*

$$\boldsymbol{x}_{t_m} = (1 - t_m)\boldsymbol{x}_0 + t_m(\boldsymbol{x}_1 + \sigma_{\min}\boldsymbol{x}_0), \quad t_m \in (0, 1), \boldsymbol{x}_0 \sim \mathcal{N}(\boldsymbol{0}, \boldsymbol{I}), \tag{8}$$

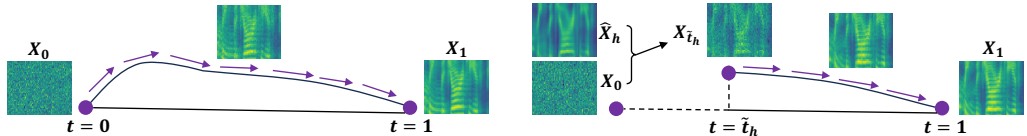

Figure 1: Inference process. Left: standard FM; Right: proposed SFM.

*we can divide the paths into two segments at $t_m$ and represent the flow and VF using piecewise functions:*

$$\boldsymbol{x}_t = \begin{cases} (1 - \frac{t}{t_m})\boldsymbol{x}_0 + \frac{t}{t_m}\boldsymbol{x}_{t_m}, & \text{if } t < t_m, \\ (1 - \frac{t-t_m}{1-t_m})\boldsymbol{x}_{t_m} + \frac{t-t_m}{1-t_m}(\boldsymbol{x}_1 + \sigma_{\min}\boldsymbol{x}_0), & \text{if } t \geq t_m, \end{cases} \tag{9}$$

$$\boldsymbol{u}_t = \begin{cases} \frac{1}{t_m}(\boldsymbol{x}_{t_m} - \boldsymbol{x}_0), & \text{if } t < t_m, \\ \frac{1}{1-t_m}(\boldsymbol{x}_1 + \sigma_{\min}\boldsymbol{x}_0 - \boldsymbol{x}_{t_m}), & \text{if } t \geq t_m. \end{cases} \tag{10}$$

### 3.2 Proposed methods on coarse-to-fine FM-based TTS

We define the mel-spectrogram of an audio waveform as $\boldsymbol{X} \in \mathbb{R}^{N \times F}$, where $N$ is the number of time frames and $F$ is the number of frequency bins (channels). Specifically, $\boldsymbol{X}^n \in \mathbb{R}^F$ denotes the $n$-th mel-spectrogram frame. We refer to the weak generator as $\boldsymbol{g}_{\boldsymbol{\omega}}$ (with learnable parameters $\boldsymbol{\omega}$), which receives text, speaker features, and other contextual information as the input condition $\boldsymbol{C}$ and outputs a coarse mel-spectrogram $\hat{\boldsymbol{X}}_{\boldsymbol{g}}$. $\hat{\boldsymbol{X}}_{\boldsymbol{g}}$ is supervised to match the target sample $\boldsymbol{X}_1$, typically using an L2 loss, though other loss functions may also be applied depending on the task:

$$\mathcal{L}_{\text{coarse}} = \mathbb{E}||\hat{\boldsymbol{X}}_{\boldsymbol{g}} - \boldsymbol{X}_1||^2. \tag{11}$$

We introduce a lightweight SFM head $\boldsymbol{h}_{\boldsymbol{\psi}}$ with learnable parameters $\boldsymbol{\psi}$ (see details in Appendix B), which takes the final hidden states $\hat{\boldsymbol{H}}_{\boldsymbol{g}}$ from $\boldsymbol{g}_{\boldsymbol{\omega}}$ as input and outputs a scaled mel-spectrogram $\hat{\boldsymbol{X}}_{\boldsymbol{h}}$. Note that $\hat{\boldsymbol{X}}_{\boldsymbol{g}}$ is obtained by applying a linear projection to $\hat{\boldsymbol{H}}_{\boldsymbol{g}}$. In addition, $\boldsymbol{h}_{\boldsymbol{\psi}}$ needs to predict a time $\hat{t}_{\boldsymbol{h}} \in (0,1)$ and an estimated variance $\hat{\sigma}_{\boldsymbol{h}}^2$ for $\hat{\boldsymbol{X}}_{\boldsymbol{h}}$:

$$\hat{\boldsymbol{H}}_{\boldsymbol{g}}, \hat{\boldsymbol{X}}_{\boldsymbol{g}} = \boldsymbol{g}_{\boldsymbol{\omega}}(\boldsymbol{C}), \quad \hat{\boldsymbol{X}}_{\boldsymbol{h}}, \hat{t}_{\boldsymbol{h}}, \hat{\sigma}_{\boldsymbol{h}}^2 = \boldsymbol{h}_{\boldsymbol{\psi}}(\hat{\boldsymbol{H}}_{\boldsymbol{g}}). \tag{12}$$

#### 3.2.1 Orthogonal projection onto CondOT paths

Since the exact location of $\hat{\boldsymbol{X}}_{\boldsymbol{h}}$ on the CondOT paths and the corresponding time $t_{\boldsymbol{h}}$ are unknown, we let the model *adaptively* determine $t_{\boldsymbol{h}}$ during training. To direct $\hat{\boldsymbol{X}}_{\boldsymbol{h}}$ to the CondOT paths, we find the *orthogonal projection* of $\hat{\boldsymbol{X}}_{\boldsymbol{h}}$ onto $\boldsymbol{X}_1$. According to Eq. (3), the projection coefficient is $t_{\boldsymbol{h}}$, which corresponds to an intermediate state on the mean path $\mu_t(\boldsymbol{X}_1)$. Then we estimate the time $t_{\boldsymbol{h}}$ and the variance $\sigma_{\boldsymbol{h}}^2$, and minimize the distance between $\hat{\boldsymbol{X}}_{\boldsymbol{h}}$ and $t_{\boldsymbol{h}}\boldsymbol{X}_1$ via the loss $\mathcal{L}_\mu$:

$$t_{\boldsymbol{h}} = \max(0, \mathbb{E}[\frac{\text{sg}[\hat{\boldsymbol{X}}_{\boldsymbol{h}}] \cdot \boldsymbol{X}_1}{\boldsymbol{X}_1 \cdot \boldsymbol{X}_1}]), \quad \sigma_{\boldsymbol{h}}^2 = \mathbb{E}||\text{sg}[\hat{\boldsymbol{X}}_{\boldsymbol{h}}] - t_{\boldsymbol{h}}\boldsymbol{X}_1||^2, \quad \mathcal{L}_\mu = \mathbb{E}||\hat{\boldsymbol{X}}_{\boldsymbol{h}} - t_{\boldsymbol{h}}\boldsymbol{X}_1||^2, \tag{13}$$

where $\text{sg}[\cdot]$ (stop gradient) is used to simplify gradient propagation. $\sigma_{\boldsymbol{h}}^2$ represents the noise scale of $\hat{\boldsymbol{X}}_{\boldsymbol{h}}$ and can be interpreted as *intrinsic noise*. Given the large number of mel-spectrogram frames, we omit the unbiased correction term in the estimation.

When $\hat{\boldsymbol{X}}_{\boldsymbol{h}} \approx t_{\boldsymbol{h}}\boldsymbol{X}_1$, we can assume that $\hat{\boldsymbol{X}}_{\boldsymbol{h}} \sim \mathcal{N}(t_{\boldsymbol{h}}\boldsymbol{X}_1, \sigma_{\boldsymbol{h}}^2\boldsymbol{I})$ and utilize Theorem 1 to construct intermediate states on the CondOT paths:

$$\Delta = \max((1 - \sigma_{\min})t_{\boldsymbol{h}} + \sigma_{\boldsymbol{h}}, 1), \quad \tilde{\boldsymbol{X}}_{\boldsymbol{h}} = \frac{1}{\Delta}\hat{\boldsymbol{X}}_{\boldsymbol{h}}, \quad \tilde{t}_{\boldsymbol{h}} = \frac{1}{\Delta}t_{\boldsymbol{h}}, \quad \tilde{\sigma}_{\boldsymbol{h}}^2 = \frac{1}{\Delta^2}\sigma_{\boldsymbol{h}}^2, \tag{14}$$

$$\boldsymbol{X}_{\tilde{t}_{\boldsymbol{h}}} = \sqrt{\max((1 - (1 - \sigma_{\min})\tilde{t}_{\boldsymbol{h}})^2 - \tilde{\sigma}_{\boldsymbol{h}}^2, 0)}\boldsymbol{X}_0 + \tilde{\boldsymbol{X}}_{\boldsymbol{h}}, \tag{15}$$

where $\boldsymbol{X}_0 \sim \mathcal{N}(\boldsymbol{0}, \boldsymbol{I})$. During the early stages of training, it is possible that $\Delta \geq 1$. In such cases, $\hat{\boldsymbol{X}}_{\boldsymbol{h}}$ is unable to lie on the CondOT path with the *external noise* $\boldsymbol{X}_0$, resulting in deterministic model

behavior. The scaling factor $\frac{1}{\Delta}$ introduced in Theorem 1 rescales and guides $\hat{X}_h$ back onto the CondOT paths until $X_0$ can be incorporated. Then we calculate the losses for the two predicted scalars:

$$\mathcal{L}_t = (\hat{t}_h - \tilde{t}_h)^2, \quad \mathcal{L}_\sigma = (\hat{\sigma}_h^2 - \tilde{\sigma}_h^2)^2. \tag{16}$$

### 3.2.2 Single-segment piecewise flow

During training and inference, the flow starts from $\tilde{t}_h$. Therefore, we employ Theorem 2 and focus only on the second segment of the paths ($t \geq \tilde{t}_h$):

$$t_{\mathcal{U}} \sim \mathcal{U}[0, 1], \quad t_{\mathcal{S}} = \mathcal{S}(t_{\mathcal{U}}), \quad t = (1 - \tilde{t}_h)t_{\mathcal{S}} + \tilde{t}_h, \tag{17}$$

$$\boldsymbol{X}_t = \left(1 - \frac{t - \tilde{t}_h}{1 - \tilde{t}_h}\right)\boldsymbol{X}_{\tilde{t}_h} + \frac{t - \tilde{t}_h}{1 - \tilde{t}_h}(\boldsymbol{X}_1 + \sigma_{\min}\boldsymbol{X}_0) \tag{18}$$

$$= (1 - t_{\mathcal{S}})\boldsymbol{X}_{\tilde{t}_h} + t_{\mathcal{S}}(\boldsymbol{X}_1 + \sigma_{\min}\boldsymbol{X}_0), \tag{19}$$

$$\boldsymbol{U}_t = \frac{1}{1 - \tilde{t}_h}(\boldsymbol{X}_1 + \sigma_{\min}\boldsymbol{X}_0 - \boldsymbol{X}_{\tilde{t}_h}), \tag{20}$$

where $\mathcal{S}$ is an arbitrary time scheduler for the randomly sampled $t$.

Finally, combined with the CFM loss, the overall loss for the SFM framework is given by:

$$\mathcal{L}_{\mathrm{CFM}} = \mathbb{E}_{t, p_t(\boldsymbol{X}_t)}||\boldsymbol{v}_{\boldsymbol{\theta}}(\boldsymbol{X}_t, t) - \boldsymbol{U}_t||^2, \quad \mathcal{L}_{\mathrm{SFM}} = \mathcal{L}_{\mathrm{coarse}} + \mathcal{L}_t + \mathcal{L}_\sigma + \mathcal{L}_\mu + \mathcal{L}_{\mathrm{CFM}}. \tag{21}$$

### 3.2.3 Inference with SFM strength

During inference, we observe that the adaptively determined $t_h$ tends to be small, which limits the amount of prior information. To address this issue, we introduce a hyperparameter called *SFM strength* $\alpha \geq 1$, which scales up the $\hat{t}_h$ to encourage stronger guidance from $\hat{X}_h$

$$\Delta = \max(\alpha((1 - \sigma_{\min})\hat{t}_h + \hat{\sigma}_h), 1), \quad \tilde{X}_h = \frac{\alpha}{\Delta}\hat{X}_h, \quad \tilde{t}_h = \frac{\alpha}{\Delta}\hat{t}_h, \quad \tilde{\sigma}_h^2 = \frac{\alpha^2}{\Delta^2}\hat{\sigma}_h^2, \tag{22}$$

Substituting $\tilde{X}_h$, $\tilde{t}_h$, and $\tilde{\sigma}_h^2$ into Eq. (15) yields $X_{\tilde{t}_h}$. We can obtain the predicted results $X_{\mathrm{pred}}$ by solving the integral $\boldsymbol{X}_{\mathrm{pred}} = \boldsymbol{X}_{\tilde{t}_h} + \int_{\tilde{t}_h}^1 \boldsymbol{v}_{\boldsymbol{\theta}}(\boldsymbol{X}_t, t)\, dt$ with an ODE solver. For each model employing the SFM method, we determine the optimal $\alpha$ on the validation set. In most cases, the optimal $\alpha$ is relatively small, resulting in $\Delta = 1$. The scaling factor $\frac{\alpha}{\Delta}$ has a theoretical upper bound of $\frac{1}{(1 - \sigma_{\min})\hat{t}_h + \hat{\sigma}_h}$.

We provide **concise algorithm boxes** in Appendix C, using minimal notation for clarity and including implementation details. An example of SFM inference is illustrated in Fig. 1.

## 4 Experimental setup

### 4.1 Datasets

We use LJ Speech [17], VCTK [44], and LibriTTS [50] in our experiments, where LJ Speech is a single-speaker dataset and the others are multi-speaker datasets. For LJ Speech and VCTK, the training, validation, and test sets are divided following the setting of Matcha-TTS, which follows VITS's settings.[1] Each validation set contains 100 utterances, and each test set contains 500 utterances.

For LibriTTS, the *train* subsets are used as the training set. We construct the validation and test sets with the *dev-clean* and *test-clean* subsets, respectively. We adopt the cross-sentence evaluation approach in CosyVoice and F5-TTS: utterances from each subset are paired to form <prompt, target> pairs, where the prompt utterance is used as the reference to TTS models and the transcript of the target utterance serves as the text input. The prompt utterances are selected such that their durations are rounded to between 3 and 4 seconds, while the duration of target utterances is between 4 and 10 seconds. We set the validation and test sets to contain 200 and 1000 utterances, respectively.

---

[1]`https://github.com/jaywalnut310/vits/tree/main/filelists`

Table 1: **Training and inference configurations of TTS models.** $\beta$: CFG strength. †: 24,000 maximum frames.

| Model | Vocoder | Dataset | Epochs | Batch Size | Learning Rate | Warmup | GPUs | $\beta$ | Solver |
|-------|---------|---------|--------|-----------|---------------|--------|------|---------|--------|
| Matcha-TTS | Vocos | LJ Speech | 800 | 128 | $4 \times 10^{-4}$ | – | 1 | – | Euler |
| Matcha-TTS | Vocos | VCTK | 800 | 128 | $2 \times 10^{-4}$ | – | 2 | – | Euler |
| StableTTS | Vocos | VCTK | 800 | 128 | $2 \times 10^{-4}$ | 10% | 2 | 3 | Dopri(5) |
| CosyVoice | HiFi-GAN | LibriTTS | 200 | dynamic† | $1 \times 10^{-4}$ | 10% | 8 | 0.7 | Euler |
| CosyVoice-DiT | Vocos | LibriTTS | 200 | 256 | $1 \times 10^{-4}$ | 10% | 4 | 3 | Dopri(5) |

## 4.2 Model implementations

Since the SFM head (the architecture is shown in Appendix B) receives hidden states as input for richer contextual information, all length regulators in each model upsample the hidden states. Several neural network layers smooth the upsampled hidden states, which are then used to generate coarse mel-spectrograms and input to the SFM head as in Eq. (12). The coarse mel-spectrograms are supervised using the target mel-spectrograms via Eq. (11).

**Matcha-TTS:** We utilize Matcha-TTS with the officially pre-trained Vocos [39] as the vocoder. Since the official implementation does not include smoothing layers for the encoder outputs, we designate the final block of the six-block encoder to serve as smoothing layers.

**StableTTS:** Although StableTTS incorporates a reference encoder to extract speaker embeddings and enables zero-shot TTS, we observe that the extracted speaker embeddings are unstable and lead to unstable encoder outputs. This instability causes the simple SFM head to struggle with converging and predicting accurate $\hat{t}_h$ and $\hat{\sigma}_h^2$. Therefore, we use ID-based speaker embeddings with random initialization instead of the reference encoder. Both the encoder and decoder are configured with six DiT blocks. The Vocos pre-trained in StableTTS is used as the vocoder.

**CosyVoice:** As discussed in Section 2.3, we only train the flow module of CosyVoice. We use the officially pre-trained speech tokenizer and LLM to generate speech tokens for training and inference, respectively. The pre-trained HiFi-GAN [20] in CosyVoice is used as the vocoder. In the original FM module, the encoder takes only speech tokens as input, while speaker embeddings and masked mel-spectrograms are used as conditions for the FM decoder. However, under this configuration, we find that the encoder fails to produce coarse mel-spectrograms with sufficient speaker features. To address this, we concatenate the speech tokens, speaker embeddings, and masked mel-spectrograms, and feed them jointly into the encoder. For ablation studies, we evaluate a variant (denoted as **SFM-t**) with the SFM method in which the encoder takes only speech tokens as input.

**CosyVoice-DiT:** As discussed in Section 2.3, we construct CosyVoice-DiT using the encoder and decoder of StableTTS, each of which contains six DiT blocks. The speech tokens are fed into the encoder, while the speaker embeddings are incorporated into every DiT block via adaLN-Zero [32]. The Vocos pre-trained in StableTTS is used as the vocoder.

**Baseline models:** We adopt and train the original models as baseline models. Due to necessary architectural modifications, no direct baseline is available for StableTTS and CosyVoice-DiT.

**Ablated models:** For each model, we construct an ablated version to enable a more direct comparison with the SFM model, mainly including the addition of coarse loss and the SFM head. In the ablated models, the SFM head only outputs the $\hat{X}_h$ as the condition for the FM module. Therefore, the only difference between the ablated and SFM models is whether the SFM method is applied. This design allows us to isolate and assess the effect of the proposed SFM method.

**SFM-c:** In our SFM method, the coarse mel-spectrograms are not used as FM conditions. For ablation studies, we also evaluate a variant (denoted as SFM-c) that not only applies the SFM method but also uses coarse mel-spectrograms as FM conditions.

## 4.3 Training and inference configurations

All training, inference, and objective evaluations are conducted on 96 GB Nvidia H100 GPUs with half precision (FP16). We follow the official configurations of each baseline as closely as possible, and some key settings are summarized in Table 1. Specifically, a warmup parameter indicates that

Table 2: Adaptive-step ODE solvers used in our experiments. All of them are variants of the explicit Runge–Kutta family.

| Abbreviation | Full Name | Order |
|---|---|---|
| Heun(2) | Adaptive Heun's Method | 2 |
| Fehlberg(2) | Runge–Kutta–Fehlberg Method | 2 |
| Bosh(3) | Bogacki–Shampine Method | 3 |
| Dopri(5) | Dormand–Prince Method | 5 |

a learning rate scheduler is used. Because of our large batch sizes, we increase the constant or peak learning rates based on the linear scaling rule, and then reduce them if gradient explosions are observed. The presence of a CFG strength indicates the application of CFG.

### 4.4 Evaluations

#### 4.4.1 Objective evaluation

We evaluate model performance using three pseudo-MOS prediction models: UTMOS [37], UTMOSv2 [3], and Distill-MOS [41]. Their average is defined as $\overline{PMOS}$ and is used for selecting the optimal $\alpha$ in SFM methods. In addition, we report word error rate (WER) and speaker similarity (SIM). For WER, we use Whisper-large-v3 [34] as the speech recognition model to transcribe speech. For SIM, we use WavLM-base-plus-sv [5, 40][2] to extract speaker embeddings and compute the cosine similarity between synthesized and ground-truth speech.

#### 4.4.2 Subjective evaluation

We conducted comparative MOS (CMOS) and similarity MOS (SMOS) tests to evaluate naturalness and similarity, respectively. In each single test, we recruited 20 native English listeners on Prolific[3] with £9 per hour.

**CMOS:** We use our proposed model with SFM as the reference system, and all other systems are compared against it. For each system, five utterances are selected and paired with corresponding utterances from the SFM model. Listeners are presented with utterance pairs and asked to rate the naturalness difference on a 7-point scale ranging from $-3$ to $+3$. A score of $+3$ indicates that the first utterance sounds much better than the second, while $-3$ indicates the opposite.

**SMOS:** SMOS tests are only conducted for cross-sentence evaluations to evaluate the similarity between the ground-truth prompt utterance and the target utterance from different systems. Each system provides five utterances with corresponding prompts. Listeners are presented with <prompt, target> utterance pairs and asked to rate how similar the target sounds to the prompt on a 5-point scale ranging from 1 to 5, where 1 indicates "not similar at all" and 5 indicates "extremely similar".

### 4.5 SFM strength selection

We determine the optimal SFM strength $\alpha$ on the corresponding validation sets. Starting from 1.0, we increment $\alpha$ by 1.0 and objectively evaluate the generated speech to identify the value that yields the highest $\overline{PMOS}$. We then examine its neighbors at $\alpha \pm 0.5$ for potential improvements. Finer-grained search is avoided to prevent overfitting and ensure the robustness of the SFM method.

### 4.6 ODE solvers and speed analysis

For evaluations, we follow the official ODE solver settings of each model. For speed analysis with adaptive-step ODE solvers, we use the *odeint* from torchdiffeq [4] with various solvers. The default relative and absolute tolerances ($1 \times 10^{-7}$ and $1 \times 10^{-9}$) are too strict, resulting in significantly longer inference time. Therefore, we adopt the settings used in StableTTS: $1 \times 10^{-5}$ for both tolerances. As

---

[2]`https://huggingface.co/microsoft/wavlm-base-plus-sv`
[3]`https://www.prolific.com`

Table 3: **Partial objective evaluation results on validation sets for $\alpha$ selection. Complete results are provided in Appendix D.** The highest value for each metric is highlighted in bold.

| $\alpha$ | $\tilde{t}_g$ | $\tilde{\sigma}_g$ | $\overline{\text{PMOS}}$↑ | UTMOS↑ | UTMOSv2↑ | Distill-MOS↑ | WER(%)↓ | SIM↑ |
|---|---|---|---|---|---|---|---|---|
| \multicolumn{9}{l}{*Matcha-TTS (SFM) trained on LJ Speech*} | | | | | | | | |
| 1.0 | 0.099 | 0.092 | 4.036 | 4.194 | 3.721 | 4.192 | 4.641 | 0.972 |
| 2.0 | 0.198 | 0.183 | 4.158 | **4.305** | 3.834 | 4.337 | 3.496 | **0.973** |
| 2.5 | 0.248 | 0.229 | **4.176** | 4.276 | **3.872** | 4.381 | 3.556 | 0.972 |
| 3.0 | 0.297 | 0.275 | 4.168 | 4.260 | 3.842 | 4.402 | 3.496 | 0.970 |
| 3.5 | 0.347 | 0.320 | 4.132 | 4.190 | 3.802 | 4.403 | 3.496 | 0.969 |
| 4.0 | 0.397 | 0.366 | 4.107 | 4.137 | 3.763 | **4.421** | 3.556 | 0.966 |
| 5.0 | 0.496 | 0.458 | 4.025 | 3.977 | 3.694 | 4.403 | 3.376 | 0.960 |
| 6.0 | 0.520 | 0.480 | 3.997 | 3.958 | 3.648 | 4.386 | **3.315** | 0.958 |
| 7.0 | 0.520 | 0.480 | 3.990 | 3.960 | 3.625 | 4.386 | **3.315** | 0.958 |
| 8.0 | 0.520 | 0.480 | 3.990 | 3.959 | 3.625 | 4.386 | **3.315** | 0.958 |
| 9.0 | 0.520 | 0.480 | 3.993 | 3.956 | 3.638 | 4.386 | **3.315** | 0.958 |
| 10.0 | 0.520 | 0.480 | 3.987 | 3.955 | 3.620 | 4.386 | **3.315** | 0.958 |

shown in Table 2, we employ Huen(2), Fehlberg(2), Bosh(3), and Dopri(5) as adaptive-step solvers, where the number in () denotes the order.

We adopt real-time factor (RTF) and number of function evaluations (NFE) as metrics for speed evaluation, where RTF measures the ratio between the ODE solver inference time and the corresponding audio duration, and NFE indicates how many times the solver queries the model. To reduce the influence of runtime variance, we choose 100 utterances from each model's validation set and run each solver five times, reporting the average RTF. Note that the NFE is constant across runs.

## 5 Experimental results and analysis

### 5.1 Evaluation results

Due to page limitations, Table 3 and Table 5 only provide partial results, and complete results are provided in Appendix D and F, respectively. Other minor analysis is provided in Appendix E.

**SFM strength selection:** In Table 3 and the tables in Appendix D, $\tilde{t}_g$ and $\tilde{\sigma}_g$ are computed according to Eq. 22, and the reported values represent their means over all utterances in the validation sets. From these tables, it is evident that using values of $\alpha > 1.0$ significantly improves both pseudo-MOS scores and WER. The $\overline{\text{PMOS}}$, which serves as the selection criterion, tends to increase initially and then decrease as $\alpha$ grows. This observation suggests that the adaptively determined $t_h$ during training is generally small. As a result, the intermediate state constructed at inference with $\alpha = 1.0$ corresponds to an early position on the CondOT paths. When Gaussian noise $X_0$ is added at this stage, the resulting signal-to-noise ratio is low, making it difficult for the flow decoder to extract sufficient information and weakening the guidance during early sampling steps.

According to Eq. (13), $\hat{X}_h$ is supervised by the linearly down-scaled $t_h X_1$. This allows us to apply the $\alpha$ during inference to linearly scale up $\tilde{X}_h$ in Eq. 22, thereby enhancing its guidance effect. However, due to estimation errors, increasing $\alpha$ also leads to a growing distance between $\tilde{X}_h$ and $\tilde{t}_h X_1$. Meanwhile, the incorporated $X_0$ decreases and results in a more deterministic sampling process. These factors ultimately reduce generation quality.

**Objective evaluations:** As shown in Table 4, all SFM models outperform their corresponding baseline and ablated models regarding pseudo-MOS scores. However, the results for WER and SIM are more variable, with improvements observed in some cases but not in all. This suggests that there is still room to improve the alignment quality of the SFM method.

**Subjective evaluations:** As shown in Table 4, SFM models achieve better performance in both CMOS and SMOS tests compared to their corresponding baseline, ablated, and SFM variants. This demonstrates that the SFM method efficiently improves the naturalness of synthesized speech.

**CosyVoice (SFM-t):** When only speech tokens are input into the encoder of the flow module, CosyVoice (SFM-t) exhibits lower SIM and significantly worse SMOS performance. This is because the CosyVoice tokenizer is pre-trained with an ASR objective to extract semantic tokens, which

Table 4: **Evaluation results on test sets.** * indicates statistically significant differences ($p < 0.05$) compared with SFM models in subjective evaluations. The highest value for each metric is bolded.

| System | UTMOS↑ | UTMOSv2↑ | Distill-MOS↑ | WER↓ | SIM↑ | CMOS↑ | SMOS↑ |
|---|---|---|---|---|---|---|---|
| ***Matcha-TTS trained on LJ Speech*** | | | | | | | |
| Ground truth | 4.380 | 3.964 | 4.241 | 3.566 | 1.000 | +0.22 | – |
| Reconstructed | 4.085 | 3.739 | 4.208 | 3.472 | 0.993 | +0.12 | – |
| Baseline | 4.186 | 3.692 | 4.282 | **3.308** | 0.971 | −0.48 | – |
| Ablated | 4.217 | 3.763 | 4.311 | 3.355 | 0.972 | −0.27 | – |
| SFM ($\alpha$=2.5) | **4.257** | **3.848** | **4.386** | 3.413 | **0.972** | 0.00 | – |
| ***Matcha-TTS trained on VCTK*** | | | | | | | |
| Ground truth | 3.999 | 3.562 | 3.986 | 1.534 | 1.000 | +0.16 | – |
| Reconstructed | 3.819 | 3.246 | 3.977 | 1.666 | 0.985 | +0.08 | – |
| Baseline | 4.008 | 2.978 | 3.870 | 1.534 | 0.939 | −0.31* | – |
| Ablated | 4.026 | 2.997 | 3.872 | 1.613 | **0.941** | −0.39* | – |
| SFM ($\alpha$=3.5) | **4.106** | **3.105** | **3.898** | **0.952** | 0.937 | 0.00 | – |
| ***StableTTS trained on VCTK*** | | | | | | | |
| Ground truth | 3.999 | 3.562 | 3.986 | 1.534 | 1.000 | +0.48* | – |
| Reconstructed | 3.360 | 2.908 | 3.855 | 1.719 | 0.972 | +0.04 | – |
| Ablated | 3.328 | 2.958 | 3.929 | 1.798 | 0.932 | −0.34* | – |
| SFM ($\alpha$=3.0) | **3.516** | **3.020** | **3.953** | **1.745** | **0.933** | 0.00 | – |
| SFM-c ($\alpha$=4.5) | 3.507 | 2.899 | 3.934 | 1.877 | 0.931 | −0.37* | – |
| ***CosyVoice trained on LibriTTS*** | | | | | | | |
| Ground truth | 4.136 | 3.262 | 4.345 | 3.180 | 1.000 | +0.19 | 3.40 |
| Reconstructed | 3.942 | 3.126 | 4.336 | 3.146 | 0.993 | −0.24 | 2.82* |
| Baseline | 4.191 | 3.303 | 4.481 | **3.513** | 0.932 | −0.21* | 3.47 |
| Ablated | 4.183 | 3.369 | 4.487 | 3.578 | **0.932** | −0.14 | 3.58 |
| SFM ($\alpha$=2.0) | **4.194** | **3.480** | 4.541 | 3.810 | 0.931 | 0.00 | **3.67** |
| SFM-t ($\alpha$=2.5) | 4.132 | 3.336 | **4.547** | 3.987 | 0.914 | −0.09 | 2.66* |
| ***CosyVoice-DiT trained on LibriTTS*** | | | | | | | |
| Ground truth | 4.136 | 3.262 | 4.345 | 3.180 | 1.000 | +0.23 | 3.31 |
| Reconstructed | 3.322 | 2.855 | 4.211 | 3.144 | 0.989 | −0.12 | 2.86* |
| Ablated | 3.499 | 3.086 | 4.316 | 3.614 | **0.936** | −0.31* | 3.15 |
| SFM ($\alpha$=2.5) | 3.751 | **3.171** | **4.502** | **3.598** | 0.932 | 0.00 | **3.21** |
| SFM-c ($\alpha$=4.0) | **3.752** | 3.156 | 4.496 | 3.634 | 0.929 | −0.06 | 3.10 |

contain limited speaker-specific information. Although speaker-related features are employed as flow conditions, they fail to guide the generated speech with sufficient speaker characteristics. This further highlights the importance of early-stage flow inference, as errors introduced at the beginning are difficult to correct in later sampling steps.

**SFM-c:** When using coarse mel-spectrograms as flow conditions, the adaptively determined $t_h$ tends to converge to 0 in models with U-Net-based architectures, rendering the SFM-c variant inapplicable to these models. Models using DiT blocks, such as StableTTS (SFM-c) and CosyVoice-DiT (SFM-c), perform worse in subjective evaluations than their corresponding SFM models. These results suggest that, for our SFM method, using coarse mel-spectrograms as flow conditions not only fails to improve performance but can also degrade synthesis quality or invalidate the method.

## 5.2 Acceleration of adaptive-step ODE solvers

Table 5 and the tables in Appendix F report only the mean RTF for clarity, as the standard deviations across the five runs are all below 0.013. We also present the speedup rate in terms of RTF, measured relative to the ablated model. From these tables, we observe that increasing $\alpha$ improves the signal-to-noise ratio of the initial state during flow inference, which stabilizes the ODE solving process. As a result, fewer forward steps are required, and the overall inference time is significantly reduced.

Table 5: **Partial RTF and NFE results for adaptive-step ODE solvers. Complete results are provided in Appendix F.** $\overline{\text{RTF}}$ denotes the mean RTF. Rate (%) denotes the relative speedup in terms of $\overline{\text{RTF}}$ compared to the ablated model.

| System | Heun(2) | | | Fehlberg(2) | | | Bosh(3) | | | Dopri(5) | | |
|---|---|---|---|---|---|---|---|---|---|---|---|---|
| | $\overline{\text{RTF}}\downarrow$ | Rate↑ | NFE↓ | $\overline{\text{RTF}}\downarrow$ | Rate↑ | NFE↓ | $\overline{\text{RTF}}\downarrow$ | Rate↑ | NFE↓ | $\overline{\text{RTF}}\downarrow$ | Rate↑ | NFE↓ |
| *Matcha-TTS trained on LJ Speech* | | | | | | | | | | | | |
| Baseline | 0.407 | -1.496 | 312.36 | 0.054 | 3.571 | 44.82 | 0.271 | -0.370 | 223.95 | 0.142 | 2.069 | 120.10 |
| Ablated | 0.401 | 0.000 | 306.72 | 0.056 | 0.000 | 45.28 | 0.270 | 0.000 | 221.81 | 0.145 | 0.000 | 121.46 |
| SFM ($\alpha$=1.0) | 0.361 | 9.858 | 277.56 | 0.053 | 4.171 | 44.08 | 0.225 | 16.924 | 186.23 | 0.132 | 9.100 | 111.14 |
| SFM ($\alpha$=2.0) | 0.299 | 25.541 | 229.43 | 0.048 | 13.306 | 39.16 | 0.187 | 30.931 | 153.08 | 0.115 | 20.484 | 96.02 |
| SFM ($\alpha$=3.0) | 0.249 | 37.931 | 191.49 | 0.043 | 22.749 | 34.88 | 0.157 | 41.895 | 129.02 | 0.100 | 30.753 | 84.20 |
| SFM ($\alpha$=4.0) | 0.207 | 48.486 | 156.78 | 0.037 | 34.246 | 29.74 | 0.131 | 51.576 | 107.90 | 0.086 | 40.559 | 72.56 |
| SFM ($\alpha$=5.0) | 0.157 | 60.825 | 120.07 | 0.027 | 50.968 | 22.14 | 0.110 | 59.266 | 90.74 | 0.076 | 47.605 | 63.74 |

Notably, this acceleration effect is limited to adaptive-step solvers, as fixed-step solvers perform a predefined number of steps, and thus cannot leverage the improved stability of the initial stages to reduce inference cost.

## 6 Related works

Our proposed SFM method extends the idea of the shallow diffusion mechanism. To the best of our knowledge, while no prior work exactly matches our approach, several studies adopt similar strategies or pursue similar objectives. As discussed in Section 3.1, ReRFlow [47] applies piecewise reflow to divided flow trajectories. PixelFlow also adopts piecewise flow for multi-scale resolution generation. In our case, we utilize piecewise flow to divide the CondOT paths into two segments and use only the last segment. In addition, shortcut models [11] employ a self-consistency mechanism to construct shortcuts along the CondOT paths. Modifying flow matching [21] samples from a Gaussian distribution centered at a coarse output instead of the standard normal distribution, and further adopts deterministic inference.

## 7 Limitations

This work applies only minimal modifications to the backbone TTS models and uses a simple SFM head to demonstrate the effectiveness of our proposed SFM method. Therefore, there is considerable room for improving the SFM framework and corresponding implementation. For example:

1. A more powerful SFM head could be used when the weak generator is unstable.

2. In our implementations, although text or semantic tokens are used to generate the intermediate states, they are not further incorporated as conditions in the FM process. Introducing explicit semantic conditioning may improve the alignment of the final outputs with the input semantic features.

3. When applying SFM on CosyVoice, we directly concatenate the speech tokens, speaker embeddings, and masked mel-spectrograms as the encoder input. This naive fusion strategy may negatively affect cross-modal alignment and calls for a more elaborate integration approach.

## 8 Conclusion

We introduce a novel Shallow Flow Matching (SFM) method for coarse-to-fine TTS models. SFM leverages coarse representations to construct intermediate states on the CondOT path, enabling FM module to produce more stable generation, more natural synthetic speech, and faster inference when using adaptive-step ODE solvers.

SFM focuses on cases where the FM module serves as a refiner. Although it is validated on TTS tasks, the underlying framework and theoretical foundation are general. It also holds potential for other domains, such as speech denoising and enhancement, as well as image generation tasks like denoising and super-resolution. Further exploration of its applications to these tasks remains a promising direction for future work.

# 9 Acknowledgements

This research was supported by JST Moonshot Grant Number JPMJMS2011 and JST SPRING Grant Number JPMJSP2108.

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

# A Theorem Proofs

**Theorem 1.** *For any random variable $x_m \sim \mathcal{N}(t_m x_1, \sigma_m^2 I)$, where $t_m \in [0, \infty)$ and $\sigma_m \in (0, \infty)$, we define a transformation that maps $x_m$ onto the CondOT paths. The output distribution varies continuously with respect to $t_m$ and $\sigma_m$ under the Wasserstein-2 metric.*

$$\Delta = (1 - \sigma_{\min})t_m + \sigma_m, \tag{23}$$

$$x_\tau = \begin{cases} \sqrt{(1 - (1 - \sigma_{\min})t_m)^2 - \sigma_m^2} x_0 + x_m, & \text{if } \Delta < 1, \\ \frac{1}{\Delta} x_m, & \text{if } \Delta \geq 1, \end{cases} \tag{24}$$

*where $x_0 \sim \mathcal{N}(0, I)$, with corresponding $\tau = \min(t_m, \frac{t_m}{\Delta})$ on the path.*

*Proof.* Since $x_m \sim \mathcal{N}(t_m x_1, \sigma_m^2 I)$ and $x_0 \sim \mathcal{N}(0, I)$ are independent, as a linear combination of them, $x_\tau$ also follows a Gaussian distribution with:

$$\tau = \begin{cases} t_m, & \text{if } \Delta < 1, \\ \frac{1}{\Delta} t_m, & \text{if } \Delta \geq 1. \end{cases} \tag{25}$$

$$\mu(x_\tau) = \begin{cases} t_m x_1 = \tau x_1, & \text{if } \Delta < 1, \\ \frac{1}{\Delta} t_m x_1 = \tau x_1, & \text{if } \Delta \geq 1, \end{cases} \tag{26}$$

$$\sigma(x_\tau) = \begin{cases} \sqrt{(1 - (1 - \sigma_{\min})t_m)^2 - \sigma_m^2 + \sigma_m^2} = 1 - (1 - \sigma_{\min})\tau, & \text{if } \Delta < 1, \\ \frac{\sigma_m}{\Delta} = 1 - (1 - \sigma_{\min})\frac{t_m}{\Delta} = 1 - (1 - \sigma_{\min})\tau, & \text{if } \Delta \geq 1, \end{cases} \tag{27}$$

where $\mu(\cdot)$ denotes the mean and $\sigma(\cdot)$ denotes the std.

Since $\mu(x_\tau)$ and $\sigma(x_\tau)$ satisfy Eq. (3), $x_\tau$ lies on the CondOT paths.

We denote the transformation as $T : [0, 1] \times (0, \infty) \times \mathbb{R}^d \times \mathbb{R}^d \to \mathbb{R}^d$. Since $x_0 \sim \mathcal{N}(0, I)$ and $x_1$ is a sample from the dataset, the distribution of $x_\tau$ depends only on $t_m$ and $\sigma_m$, conditioned on $x_1$. Therefore, we denote the distribution of $x_\tau$ as:

$$x_\tau \sim \mathcal{N}(\mu(x_\tau), \sigma(x_\tau)^2 I) = T_{x_1}(t_m, \sigma_m). \tag{28}$$

To verify that $T_{x_1}(t_m, \sigma_m)$ is continuous in the Wasserstein-2 metric with respect to $t_m$ and $\sigma_m$, we use formula:

$$W_2(\mathcal{N}_1(\mu_1, \Sigma_1), \mathcal{N}_2(\mu_2, \Sigma_2)) = \sqrt{\|\mu_1 - \mu_2\|^2 + \text{Tr}\left(\Sigma_1 + \Sigma_2 - 2(\Sigma_1^{1/2}\Sigma_2\Sigma_1^{1/2})^{1/2}\right)}. \tag{29}$$

If our distributions are isotropic, we have:

$$W_2(\mathcal{N}_1(\mu_1, \sigma_1^2 I), \mathcal{N}_2(\mu_2, \sigma_2^2 I)) = \sqrt{\|\mu_1 - \mu_2\|^2 + n(\sigma_1 - \sigma_2)^2}, \tag{30}$$

where $n$ is the dimension of the variables. We define $\delta > 0$ for proof.

For $t_m$:

I. When $t_m < \frac{1 - \sigma_m}{1 - \sigma_{\min}}$, then $\Delta < 1$:

$$\lim_{\delta \to 0} W_2(T_{x_1}(t_m - \delta, \sigma_m), T_{x_1}(t_m, \sigma_m)) = \lim_{\delta \to 0} \sqrt{\|\delta x_1\|^2 + n(\delta(1 - \sigma_{\min}))^2} \tag{31}$$

$$= 0, \tag{32}$$

$$\lim_{\delta \to 0} W_2(T_{x_1}(t_m, \sigma_m), T_{x_1}(t_m + \delta, \sigma_m)) = \lim_{\delta \to 0} \sqrt{\|\delta x_1\|^2 + n(\delta(1 - \sigma_{\min}))^2} \tag{33}$$

$$= 0. \tag{34}$$

II. When $t_m = \frac{1 - \sigma_m}{1 - \sigma_{\min}}$, then $\Delta = 1$:

$$\lim_{\delta \to 0} W_2(T_{\boldsymbol{x_1}}(t_m - \delta, \sigma_m), T_{\boldsymbol{x_1}}(t_m, \sigma_m)) = \lim_{\delta \to 0} \sqrt{\|\delta \boldsymbol{x_1}\|^2 + n(\delta(1 - \sigma_{\min}))^2} \qquad (35)$$
$$= 0, \qquad (36)$$

$$\lim_{\delta \to 0} W_2(T_{\boldsymbol{x_1}}(t_m, \sigma_m), T_{\boldsymbol{x_1}}(t_m + \delta, \sigma_m)) \qquad (37)$$
$$= \lim_{\delta \to 0} \sqrt{\|[t_m - \frac{t_m + \delta}{1 + \delta(1 - \sigma_{\min})}]\boldsymbol{x_1}\|^2 + n(\sigma_m - \frac{\sigma_m}{1 + \delta(1 - \sigma_{\min})})^2} \qquad (38)$$
$$= 0. \qquad (39)$$

III. When $t_m > \frac{1 - \sigma_m}{1 - \sigma_{\min}}$, then $\Delta > 1$:

$$\lim_{\delta \to 0} W_2(T_{\boldsymbol{x_1}}(t_m - \delta, \sigma_m), T_{\boldsymbol{x_1}}(t_m, \sigma_m)) \qquad (40)$$
$$= \lim_{\delta \to 0} \sqrt{\|[\frac{t_m - \delta}{\Delta - \delta(1 - \sigma_{\min})} - \frac{t_m}{\Delta}]\boldsymbol{x_1}\|^2 + n(\frac{\sigma_m}{\Delta - \delta(1 - \sigma_{\min})} - \frac{\sigma_m}{\Delta})^2} \qquad (41)$$
$$= 0, \qquad (42)$$

$$\lim_{\delta \to 0} W_2(T_{\boldsymbol{x_1}}(t_m, \sigma_m), T_{\boldsymbol{x_1}}(t_m + \delta, \sigma_m)) \qquad (43)$$
$$= \lim_{\delta \to 0} \sqrt{\|[\frac{t_m}{\Delta} - \frac{t_m + \delta}{\Delta + \delta(1 - \sigma_{\min})}]\boldsymbol{x_1}\|^2 + n(\frac{\sigma_m}{\Delta} - \frac{\sigma_m}{\Delta + \delta(1 - \sigma_{\min})})^2} \qquad (44)$$
$$= 0. \qquad (45)$$

For $\sigma_m$:
I. When $\sigma_m < 1 - (1 - \sigma_{\min})t_m$, then $\Delta < 1$:

$$\lim_{\delta \to 0} W_2(T_{\boldsymbol{x_1}}(t_m, \sigma_m - \delta), T_{\boldsymbol{x_1}}(t_m, \sigma_m)) = \lim_{\delta \to 0} \sqrt{\|0\boldsymbol{x_1}\|^2 + n(0))^2} \qquad (46)$$
$$= 0, \qquad (47)$$

$$\lim_{\delta \to 0} W_2(T_{\boldsymbol{x_1}}(t_m, \sigma_m), T_{\boldsymbol{x_1}}(t_m, \sigma_m + \delta)) = \lim_{\delta \to 0} \sqrt{\|0\boldsymbol{x_1}\|^2 + n(0))^2} \qquad (48)$$
$$= 0. \qquad (49)$$

II. When $\sigma_m = 1 - (1 - \sigma_{\min})t_m$, then $\Delta = 1$:

$$\lim_{\delta \to 0} W_2(T_{\boldsymbol{x_1}}(t_m, \sigma_m - \delta), T_{\boldsymbol{x_1}}(t_m, \sigma_m)) = \lim_{\delta \to 0} \sqrt{\|0\boldsymbol{x_1}\|^2 + n(0))^2} \qquad (50)$$
$$= 0, \qquad (51)$$

$$\lim_{\delta \to 0} W_2(T_{\boldsymbol{x_1}}(t_m, \sigma_m), T_{\boldsymbol{x_1}}(t_m, \sigma_m + \delta)) \qquad (52)$$
$$= \lim_{\delta \to 0} \sqrt{\|[t_m - \frac{t_m}{1 + \delta}]\boldsymbol{x_1}\|^2 + n(\sigma_m - \frac{\sigma_m + \delta}{1 + \delta})^2} \qquad (53)$$
$$= 0. \qquad (54)$$

III. When $\sigma_m > 1 - (1 - \sigma_{\min})t_m$, then $\Delta > 1$:

$$\lim_{\delta \to 0} W_2(T_{\boldsymbol{x}_1}(t_m, \sigma_m - \delta), T_{\boldsymbol{x}_1}(t_m, \sigma_m)) \tag{55}$$

$$= \lim_{\delta \to 0} \sqrt{\|[\frac{t_m}{\Delta - \delta} - \frac{t_m}{\Delta}]\boldsymbol{x}_1\|^2 + n(\frac{\sigma_m - \delta}{\Delta - \delta} - \frac{\sigma_m}{\Delta})^2} \tag{56}$$

$$= 0, \tag{57}$$

$$\lim_{\delta \to 0} W_2(T_{\boldsymbol{x}_1}(t_m, \sigma_m), T_{\boldsymbol{x}_1}(t_m, \sigma_m + \delta)) \tag{58}$$

$$= \lim_{\delta \to 0} \sqrt{\|[\frac{t_m}{\Delta} - \frac{t_m}{\Delta + \delta}]\boldsymbol{x}_1\|^2 + n(\frac{\sigma_m}{\Delta} - \frac{\sigma_m + \delta}{\Delta + \delta})^2} \tag{59}$$

$$= 0. \tag{60}$$

Thus, $T_{\boldsymbol{x}_1}(t_m, \sigma_m)$ is continuous with respect to $t_m$ and $\sigma_m$ in the Wasserstein-2 metric. $\qquad \square$

**Theorem 2.** *For arbitrary intermediate states on the conditional OT (CondOT) paths:*

$$\boldsymbol{x}_{t_m} = (1 - t_m)\boldsymbol{x}_0 + t_m(\boldsymbol{x}_1 + \sigma_{\min}\boldsymbol{x}_0), \quad t_m \in (0, 1), \boldsymbol{x}_0 \sim \mathcal{N}(\boldsymbol{0}, \boldsymbol{I}), \tag{61}$$

*we can divide the paths into two segments at $t_m$ and represent the flow and VF using piecewise functions:*

$$\boldsymbol{x}_t = \begin{cases} (1 - \frac{t}{t_m})\boldsymbol{x}_0 + \frac{t}{t_m}\boldsymbol{x}_{t_m}, & \text{if } t < t_m, \\ (1 - \frac{t - t_m}{1 - t_m})\boldsymbol{x}_{t_m} + \frac{t - t_m}{1 - t_m}(\boldsymbol{x}_1 + \sigma_{\min}\boldsymbol{x}_0), & \text{if } t \geq t_m, \end{cases} \tag{62}$$

$$\boldsymbol{u}_t = \begin{cases} \frac{1}{t_m}(\boldsymbol{x}_{t_m} - \boldsymbol{x}_0), & \text{if } t < t_m, \\ \frac{1}{1 - t_m}(\boldsymbol{x}_1 + \sigma_{\min}\boldsymbol{x}_0 - \boldsymbol{x}_{t_m}), & \text{if } t \geq t_m. \end{cases} \tag{63}$$

In the first segment ($t < t_m$) of the two-segment piecewise flow, the paths start from $\boldsymbol{x}_0$ with $\boldsymbol{x}_{t_m}$ as the target. In the second segment ($t \geq t_m$), the paths start from $\boldsymbol{x}_{t_m}$ and move towards the target $(\boldsymbol{x}_1 + \sigma_{\min}\boldsymbol{x}_0)$.

*Proof.* Substituting Eq. (61) into Eq. (62) and Eq. (63), we derive the following:

I. When $t < t_m$:

$$\boldsymbol{x}_t = (1 - \frac{t}{t_m})\boldsymbol{x}_0 + \frac{t}{t_m}\boldsymbol{x}_{t_m} \tag{64}$$

$$= (1 - \frac{t}{t_m})\boldsymbol{x}_0 + \frac{t}{t_m}(1 - t_m)\boldsymbol{x}_0 + t(\boldsymbol{x}_1 + \sigma_{\min}\boldsymbol{x}_0) \tag{65}$$

$$= (1 - t)\boldsymbol{x}_0 + t(\boldsymbol{x}_1 + \sigma_{\min}\boldsymbol{x}_0), \tag{66}$$

$$\boldsymbol{u}_t = \frac{1}{t_m}(\boldsymbol{x}_{t_m} - \boldsymbol{x}_0) \tag{67}$$

$$= \frac{1}{t_m}(t_m\boldsymbol{x}_0 + t_m(\boldsymbol{x}_1 + \sigma_{\min}\boldsymbol{x}_0)) \tag{68}$$

$$= (\boldsymbol{x}_1 + \sigma_{\min}\boldsymbol{x}_0) - \boldsymbol{x}_0. \tag{69}$$

II. When $t \geq t_m$:

$$\boldsymbol{x}_t = (1 - \frac{t - t_m}{1 - t_m})\boldsymbol{x}_{t_m} + \frac{t - t_m}{1 - t_m}(\boldsymbol{x}_1 + \sigma_{\min}\boldsymbol{x}_0) \tag{70}$$

$$= (1 - t)\boldsymbol{x}_0 + \frac{1 - t}{1 - t_m}t_m(\boldsymbol{x}_1 + \sigma_{\min}\boldsymbol{x}_0) + \frac{t - t_m}{1 - t_m}(\boldsymbol{x}_1 + \sigma_{\min}\boldsymbol{x}_0) \tag{71}$$

$$= (1 - t)\boldsymbol{x}_0 + t(\boldsymbol{x}_1 + \sigma_{\min}\boldsymbol{x}_0), \tag{72}$$

$$\boldsymbol{u}_t = \frac{1}{1 - t_m}(\boldsymbol{x}_1 + \sigma_{\min}\boldsymbol{x}_0 - \boldsymbol{x}_{t_m}) \tag{73}$$

$$= \frac{1}{1 - t_m}(\boldsymbol{x}_1 + \sigma_{\min}\boldsymbol{x}_0) - \boldsymbol{x}_0 - \frac{t_m}{1 - t_m}(\boldsymbol{x}_1 + \sigma_{\min}\boldsymbol{x}_0) \tag{74}$$

$$= (\boldsymbol{x}_1 + \sigma_{\min}\boldsymbol{x}_0) - \boldsymbol{x}_0. \tag{75}$$

Therefore, the piecewise flow and VF above satisfy Eq. (4). □

## B  SFM head

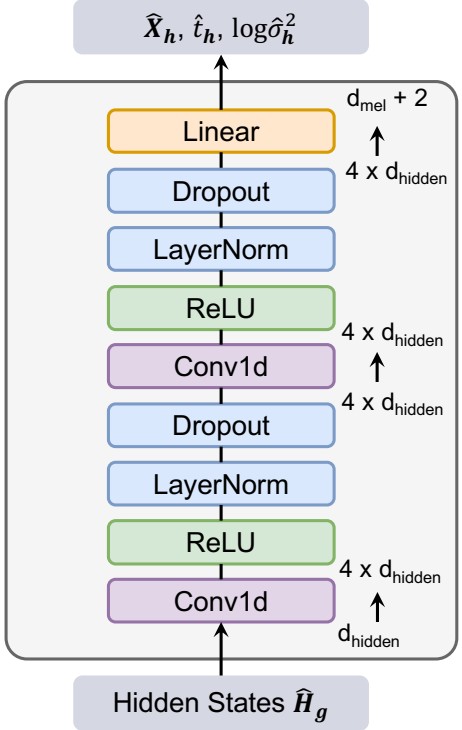

Figure 2: SFM head architecture.

As shown in Fig. 2, we use a simple and lightweight SFM head, whose architecture is derived from the *duration predictor* of VITS[4] and Matcha-TTS.[5] The kernel size and padding of the two Conv1d layers are set to 3 and 1, respectively.

The output of the SFM head can be split into $\hat{X}_h$, $\hat{t}_h$, and $\log \hat{\sigma}_h^2$, with shapes `[batch, mel, sequence]`, `[batch, 1, sequence]`, and `[batch, 1, sequence]`, respectively. Since $\hat{t}_h > 0$, we apply a Sigmoid activation to it and compute the mean over the frames axis, resulting in an output of shape `[batch, 1]`. As described in Appendix C, the SFM head predicts $\log \hat{\sigma}_h^2$ instead of $\hat{\sigma}_h^2$. $\log \hat{\sigma}_h^2$ is also averaged across frames to produce an output of shape `[batch, 1]`.

---

[4]`https://github.com/jaywalnut310/vits/blob/2e561ba58618d021b5b8323d3765880f7e0ecfdb/models.py#L98`

[5]`https://github.com/shivammehta25/Matcha-TTS/blob/108906c603fad5055f2649b3fd71d2bbdf222eac/matcha/models/components/text_encoder.py#L70`

# C Algorithms

In the practical implementation, we apply several detailed modifications and tricks:

1. The SFM-head $h_\psi$ predicts $\log \hat{\sigma}_h^2$ instead of $\hat{\sigma}_h^2$ directly to ensure numerical stability and positive outputs.

2. Due to the $\mathcal{L}_{\text{coarse}}$, the $t_h$ obtained from $X_h$ are generally larger than 0 during training. Therefore, we remove the constraint $t_h \geq 0$ in Equation 13.

3. We also modify the placement of the $\mathcal{L}_\mu$ for implementation convenience. Compared to the formulation in Equation 13, this modification results in a scaled version of the original loss by a constant factor of $\frac{1}{\Delta^2}$.

---

**Algorithm 1** Training procedure of SFM

---

**Input:** the training set $(\mathcal{X}, \mathcal{C})$.

1: **repeat**
2:     Sample $(X_1, C)$ from $(\mathcal{X}, \mathcal{C})$;
3:     Generate coarse representations

$$H_g, X_g \leftarrow g_\omega(C)$$

$$X_h, \hat{t}_h, \log \hat{\sigma}_h^2 \leftarrow h_\psi(H_g)$$

4:     Compute coarse loss

$$\mathcal{L}_{\text{coarse}} = \mathbb{E}\|X_g - X_1\|^2$$

5:     Project onto CondOT paths

$$t_h \leftarrow \mathbb{E}\left[\frac{sg[X_h] \cdot X_1}{X_1 \cdot X_1}\right], \quad \sigma_h^2 \leftarrow \mathbb{E}\|sg[X_h] - t_h X_1\|^2$$

6:     Construct the intermediate state

$$\Delta \leftarrow \max((1 - \sigma_{\min})t_h + \sigma_h, 1)$$

$$X_h \leftarrow \frac{1}{\Delta}X_h, \quad t_h \leftarrow \frac{1}{\Delta}t_h, \quad \sigma_h^2 \leftarrow \frac{1}{\Delta^2}\sigma_h^2$$

$$X_0 \sim \mathcal{N}(0, I)$$

$$X_h \leftarrow \sqrt{\max((1 - (1 - \sigma_{\min})t_h)^2 - \sigma_h^2, 0)}X_0 + X_h$$

$$\mathcal{L}_t = (\hat{t}_h - t_h)^2, \quad \mathcal{L}_\sigma = (\log \hat{\sigma}_h^2 - \log \sigma_h^2)^2, \quad \mathcal{L}_\mu = \mathbb{E}\|X_h - t_h X_1\|^2$$

7:     Apply FM

$$t \sim \mathcal{U}[0, 1], \quad t \leftarrow \mathcal{S}(t)$$

$$X_t \leftarrow (1 - t)X_h + t(X_1 + \sigma_{\min}X_0)$$

$$U_t \leftarrow \frac{1}{1 - t_h}(X_1 + \sigma_{\min}X_0 - X_h)$$

$$t \leftarrow (1 - t_h)t + t_h$$

$$\mathcal{L}_{\text{CFM}} = \mathbb{E}\|v_\theta(X_t, t) - U_t\|^2$$

8:     Apply gradient descent to minimize $\mathcal{L}_{\text{SFM}}$

$$\mathcal{L}_{\text{SFM}} = \mathcal{L}_{\text{coarse}} + \mathcal{L}_t + \mathcal{L}_\sigma + \mathcal{L}_\mu + \mathcal{L}_{\text{CFM}}$$

9: **until** convergence

---

**Algorithm 2** Inference procedure of SFM

**Input:** Condition $C$.

1: Generate coarse representations

$$\boldsymbol{H_g}, \boldsymbol{X_g} \leftarrow \boldsymbol{g_\omega}(\boldsymbol{C})$$

$$\boldsymbol{X_h}, t_{\boldsymbol{h}}, \log \sigma_{\boldsymbol{h}}^2 \leftarrow \boldsymbol{h_\psi}(\boldsymbol{H_g})$$

$$\sigma_{\boldsymbol{h}}^2 \leftarrow \exp(\log \sigma_{\boldsymbol{h}}^2)$$

2: Construct the intermediate state with SFM strength

$$\Delta \leftarrow \max(\alpha((1 - \sigma_{\min})t_{\boldsymbol{h}} + \sigma_{\boldsymbol{h}}), 1)$$

$$\boldsymbol{X_h} \leftarrow \frac{\alpha}{\Delta} \boldsymbol{X_h}, \quad t_{\boldsymbol{h}} \leftarrow \frac{\alpha}{\Delta} t_{\boldsymbol{h}}, \quad \sigma_{\boldsymbol{h}}^2 \leftarrow \frac{\alpha^2}{\Delta^2} \sigma_{\boldsymbol{h}}^2$$

$$\boldsymbol{X}_0 \sim \mathcal{N}(\boldsymbol{0}, \boldsymbol{I})$$

$$\boldsymbol{X_h} \leftarrow \sqrt{\max((1 - (1 - \sigma_{\min})t_{\boldsymbol{h}})^2 - \sigma_{\boldsymbol{h}}^2, 0)} \boldsymbol{X}_0 + \boldsymbol{X_h},$$

3: Use an ODE solver to solve the integral

$$\boldsymbol{X}_{\text{pred}} = \boldsymbol{X_h} + \int_{t_{\boldsymbol{h}}}^1 \boldsymbol{v_\theta}(\boldsymbol{X}_t, t)\, dt$$

# D Complete results of $\alpha$ selection

We provide complete results in optimal $\alpha$ selection for all models in this section, including Table 6, Table 7, Table 8, and Table 9.

Table 6: Objective evaluation results on validation sets for $\alpha$ selection (Matcha-TTS).

| $\alpha$ | $\tilde{t}_g$ | $\tilde{\sigma}_g$ | PMOS↑ | UTMOS↑ | UTMOSv2↑ | Distill-MOS↑ | WER(%)↓ | SIM↑ |
|---|---|---|---|---|---|---|---|---|
| *Matcha-TTS (SFM) trained on LJ Speech* | | | | | | | | |
| 1.0 | 0.099 | 0.092 | 4.036 | 4.194 | 3.721 | 4.192 | 4.641 | 0.972 |
| 2.0 | 0.198 | 0.183 | 4.158 | **4.305** | 3.834 | 4.337 | 3.496 | **0.973** |
| 2.5 | 0.248 | 0.229 | **4.176** | 4.276 | **3.872** | 4.381 | 3.556 | 0.972 |
| 3.0 | 0.297 | 0.275 | 4.168 | 4.260 | 3.842 | 4.402 | 3.496 | 0.970 |
| 3.5 | 0.347 | 0.320 | 4.132 | 4.190 | 3.802 | 4.403 | 3.496 | 0.969 |
| 4.0 | 0.397 | 0.366 | 4.107 | 4.137 | 3.763 | **4.421** | 3.556 | 0.966 |
| 5.0 | 0.496 | 0.458 | 4.025 | 3.977 | 3.694 | 4.403 | 3.376 | 0.960 |
| 6.0 | 0.520 | 0.480 | 3.997 | 3.958 | 3.648 | 4.386 | **3.315** | 0.958 |
| 7.0 | 0.520 | 0.480 | 3.990 | 3.960 | 3.625 | 4.386 | **3.315** | 0.958 |
| 8.0 | 0.520 | 0.480 | 3.990 | 3.959 | 3.625 | 4.386 | **3.315** | 0.958 |
| 9.0 | 0.520 | 0.480 | 3.993 | 3.956 | 3.638 | 4.386 | **3.315** | 0.958 |
| 10.0 | 0.520 | 0.480 | 3.987 | 3.955 | 3.620 | 4.386 | **3.315** | 0.958 |
| *Matcha-TTS (SFM) trained on VCTK* | | | | | | | | |
| 1.0 | 0.100 | 0.078 | 3.462 | 3.876 | 2.723 | 3.787 | 4.898 | 0.923 |
| 2.0 | 0.201 | 0.155 | 3.647 | 4.061 | 2.988 | 3.891 | 2.177 | 0.940 |
| 2.5 | 0.251 | 0.194 | 3.652 | 4.045 | 3.007 | 3.904 | 2.041 | **0.940** |
| 3.0 | 0.301 | 0.233 | 3.678 | 4.079 | 3.041 | **3.914** | 1.497 | 0.939 |
| 3.5 | 0.351 | 0.272 | **3.679** | 4.074 | **3.053** | 3.910 | 1.224 | 0.936 |
| 4.0 | 0.401 | 0.311 | 3.660 | 4.079 | 3.001 | 3.900 | **1.088** | 0.934 |
| 5.0 | 0.502 | 0.389 | 3.615 | 4.059 | 2.916 | 3.869 | **1.088** | 0.929 |
| 6.0 | 0.564 | 0.436 | 3.555 | 4.084 | 2.757 | 3.824 | 1.224 | 0.923 |
| 7.0 | 0.564 | 0.436 | 3.554 | **4.084** | 2.752 | 3.824 | 1.224 | 0.923 |
| 8.0 | 0.564 | 0.436 | 3.563 | 4.084 | 2.782 | 3.824 | 1.224 | 0.923 |
| 9.0 | 0.564 | 0.436 | 3.559 | 4.082 | 2.770 | 3.825 | 1.224 | 0.923 |
| 10.0 | 0.564 | 0.436 | 3.546 | 4.083 | 2.730 | 3.824 | 1.224 | 0.923 |

Table 7: Objective evaluation results on validation sets for $\alpha$ selection (StableTTS).

| $\alpha$ | $\tilde{t}_g$ | $\tilde{\sigma}_g$ | $\overline{\text{PMOS}}\uparrow$ | UTMOS↑ | UTMOSv2↑ | Distill-MOS↑ | WER(%)↓ | SIM↑ |
|---|---|---|---|---|---|---|---|---|
| *StableTTS (SFM) trained on VCTK* | | | | | | | | |
| 1.0 | 0.153 | 0.098 | 3.281 | 3.184 | 2.782 | 3.877 | 8.707 | 0.910 |
| 2.0 | 0.306 | 0.197 | 3.441 | 3.403 | 2.981 | 3.938 | 2.041 | 0.931 |
| 2.5 | 0.382 | 0.246 | 3.476 | 3.469 | **3.013** | **3.945** | 1.361 | 0.933 |
| 3.0 | 0.459 | 0.295 | **3.486** | 3.522 | 3.001 | 3.936 | **1.224** | **0.933** |
| 3.5 | 0.535 | 0.345 | 3.475 | 3.543 | 2.959 | 3.923 | **1.224** | 0.933 |
| 4.0 | 0.603 | 0.388 | 3.437 | 3.547 | 2.857 | 3.906 | 1.361 | 0.933 |
| 5.0 | 0.609 | 0.391 | 3.435 | 3.560 | 2.829 | 3.914 | 1.361 | 0.933 |
| 6.0 | 0.609 | 0.391 | 3.434 | **3.561** | 2.825 | 3.915 | 1.361 | 0.933 |
| 7.0 | 0.609 | 0.391 | 3.431 | 3.558 | 2.819 | 3.915 | 1.361 | 0.933 |
| 8.0 | 0.609 | 0.391 | 3.434 | 3.559 | 2.828 | 3.915 | 1.361 | 0.933 |
| 9.0 | 0.609 | 0.391 | 3.435 | 3.560 | 2.832 | 3.914 | 1.361 | 0.933 |
| 10.0 | 0.609 | 0.391 | 3.433 | 3.560 | 2.824 | 3.914 | 1.361 | 0.933 |
| *StableTTS (SFM-c) trained on VCTK* | | | | | | | | |
| 1.0 | 0.111 | 0.067 | 3.261 | 3.165 | 2.759 | 3.859 | 1.224 | 0.918 |
| 2.0 | 0.222 | 0.134 | 3.313 | 3.260 | 2.787 | 3.893 | 1.633 | 0.922 |
| 3.0 | 0.333 | 0.201 | 3.403 | 3.375 | 2.913 | 3.923 | 1.497 | 0.927 |
| 3.5 | 0.388 | 0.235 | 3.424 | 3.419 | 2.916 | **3.936** | 1.361 | 0.928 |
| 4.0 | 0.444 | 0.268 | 3.427 | 3.462 | 2.889 | 3.931 | **1.088** | 0.930 |
| 4.5 | 0.499 | 0.302 | **3.449** | 3.495 | **2.924** | 3.929 | **1.088** | 0.931 |
| 5.0 | 0.555 | 0.335 | 3.408 | 3.510 | 2.799 | 3.913 | 1.361 | 0.931 |
| 6.0 | 0.624 | 0.376 | 3.377 | 3.553 | 2.672 | 3.906 | 1.224 | 0.934 |
| 7.0 | 0.624 | 0.376 | 3.389 | 3.554 | 2.707 | 3.906 | 1.224 | 0.934 |
| 8.0 | 0.624 | 0.376 | 3.398 | **3.554** | 2.734 | 3.907 | 1.224 | 0.934 |
| 9.0 | 0.624 | 0.376 | 3.387 | 3.553 | 2.702 | 3.907 | 1.224 | **0.934** |
| 10.0 | 0.624 | 0.376 | 3.378 | 3.554 | 2.674 | 3.906 | 1.224 | 0.934 |

Table 8: Objective evaluation results on validation sets for $\alpha$ selection (CosyVoice).

| $\alpha$ | $\tilde{t}_g$ | $\tilde{\sigma}_g$ | $\overline{\text{PMOS}}\uparrow$ | UTMOS↑ | UTMOSv2↑ | Distill-MOS↑ | WER(%)↓ | SIM↑ |
|---|---|---|---|---|---|---|---|---|
| *CosyVoice (SFM) trained on LibriTTS* | | | | | | | | |
| 1.0 | 0.152 | 0.126 | 3.721 | 3.729 | 3.102 | 4.333 | 8.902 | 0.923 |
| 1.5 | 0.228 | 0.189 | 4.019 | 4.113 | 3.449 | 4.494 | 4.810 | **0.932** |
| 2.0 | 0.303 | 0.253 | **4.087** | **4.180** | **3.530** | 4.553 | 4.499 | 0.931 |
| 2.5 | 0.379 | 0.316 | 4.080 | 4.165 | 3.516 | **4.558** | 4.475 | 0.928 |
| 3.0 | 0.455 | 0.379 | 4.019 | 4.119 | 3.409 | 4.527 | 4.523 | 0.922 |
| 4.0 | 0.546 | 0.454 | 3.917 | 4.075 | 3.235 | 4.440 | **4.427** | 0.916 |
| 5.0 | 0.546 | 0.454 | 3.911 | 4.081 | 3.209 | 4.444 | 4.475 | 0.916 |
| 6.0 | 0.546 | 0.454 | 3.904 | 4.070 | 3.199 | 4.443 | 4.523 | 0.916 |
| 7.0 | 0.546 | 0.454 | 3.905 | 4.070 | 3.201 | 4.442 | 4.475 | 0.916 |
| 8.0 | 0.546 | 0.454 | 3.912 | 4.075 | 3.220 | 4.440 | **4.427** | 0.916 |
| 9.0 | 0.546 | 0.454 | 3.919 | 4.077 | 3.239 | 4.441 | 4.475 | 0.916 |
| 10.0 | 0.546 | 0.454 | 3.911 | 4.081 | 3.207 | 4.444 | 4.475 | 0.916 |
| *CosyVoice (SFM-t) trained on LibriTTS* | | | | | | | | |
| 1.0 | 0.088 | 0.152 | 3.567 | 3.501 | 2.957 | 4.242 | 14.932 | 0.917 |
| 1.5 | 0.132 | 0.228 | 3.872 | 3.932 | 3.267 | 4.417 | 6.102 | **0.928** |
| 2.0 | 0.176 | 0.304 | 4.000 | **4.089** | 3.392 | 4.520 | 4.882 | 0.924 |
| 2.5 | 0.219 | 0.380 | **4.005** | 4.078 | **3.395** | **4.543** | 4.834 | 0.911 |
| 3.0 | 0.263 | 0.456 | 3.877 | 3.940 | 3.213 | 4.477 | 4.355 | 0.896 |
| 4.0 | 0.349 | 0.605 | 3.395 | 3.498 | 2.599 | 4.087 | 4.355 | 0.868 |
| 5.0 | 0.364 | 0.635 | 3.251 | 3.375 | 2.421 | 3.955 | 4.379 | 0.865 |
| 6.0 | 0.364 | 0.636 | 3.260 | 3.376 | 2.441 | 3.963 | **4.331** | 0.865 |
| 7.0 | 0.364 | 0.636 | 3.261 | 3.380 | 2.450 | 3.953 | 4.355 | 0.864 |
| 8.0 | 0.364 | 0.636 | 3.271 | 3.379 | 2.470 | 3.963 | 4.403 | 0.865 |
| 9.0 | 0.364 | 0.636 | 3.271 | 3.377 | 2.478 | 3.958 | 4.427 | 0.865 |
| 10.0 | 0.364 | 0.636 | 3.255 | 3.375 | 2.433 | 3.955 | 4.379 | 0.865 |

Table 9: Objective evaluation results on validation sets for $\alpha$ selection (CosyVoice-DiT).

| $\alpha$ | $\tilde{t}_g$ | $\tilde{\sigma}_g$ | PMOS↑ | UTMOS↑ | UTMOSv2↑ | Distill-MOS↑ | WER(%)↓ | SIM↑ |
|---|---|---|---|---|---|---|---|---|
| *CosyVoice-DiT (SFM) trained on LibriTTS* | | | | | | | | |
| 1.0 | 0.143 | 0.120 | 3.405 | 3.077 | 2.880 | 4.257 | 8.423 | 0.924 |
| 2.0 | 0.286 | 0.239 | 3.750 | 3.569 | 3.189 | 4.493 | 4.499 | **0.934** |
| 2.5 | 0.358 | 0.299 | **3.823** | 3.694 | **3.247** | **4.529** | 4.331 | 0.933 |
| 3.0 | 0.429 | 0.359 | 3.810 | 3.720 | 3.184 | 4.528 | 4.212 | 0.929 |
| 3.5 | 0.501 | 0.419 | 3.780 | **3.758** | 3.089 | 4.493 | **4.068** | 0.925 |
| 4.0 | 0.545 | 0.455 | 3.721 | 3.750 | 2.968 | 4.444 | 4.092 | 0.922 |
| 5.0 | 0.545 | 0.455 | 3.722 | 3.750 | 2.972 | 4.444 | 4.092 | 0.922 |
| 6.0 | 0.545 | 0.455 | 3.720 | 3.749 | 2.967 | 4.444 | 4.092 | 0.922 |
| 7.0 | 0.545 | 0.455 | 3.723 | 3.751 | 2.973 | 4.445 | 4.092 | 0.922 |
| 8.0 | 0.545 | 0.455 | 3.718 | 3.750 | 2.959 | 4.444 | 4.092 | 0.922 |
| 9.0 | 0.545 | 0.455 | 3.724 | 3.750 | 2.979 | 4.444 | 4.092 | 0.922 |
| 10.0 | 0.545 | 0.455 | 3.719 | 3.750 | 2.964 | 4.444 | 4.092 | 0.922 |
| *CosyVoice-DiT (SFM-c) trained on LibriTTS* | | | | | | | | |
| 1.0 | 0.106 | 0.082 | 3.419 | 3.154 | 2.943 | 4.161 | 4.666 | 0.933 |
| 2.0 | 0.212 | 0.164 | 3.655 | 3.408 | 3.146 | 4.409 | 4.379 | **0.937** |
| 3.0 | 0.317 | 0.246 | 3.766 | 3.573 | 3.217 | 4.509 | 4.307 | 0.936 |
| 3.5 | 0.370 | 0.287 | 3.799 | 3.640 | **3.230** | **4.526** | 4.283 | 0.934 |
| 4.0 | 0.423 | 0.329 | **3.815** | 3.696 | 3.227 | 4.522 | 4.283 | 0.933 |
| 4.5 | 0.476 | 0.370 | 3.814 | 3.739 | 3.180 | 4.521 | 4.331 | 0.931 |
| 5.0 | 0.529 | 0.411 | 3.794 | 3.789 | 3.080 | 4.513 | 4.164 | 0.929 |
| 6.0 | 0.563 | 0.437 | 3.780 | **3.795** | 3.056 | 4.488 | **4.116** | 0.927 |
| 7.0 | 0.563 | 0.437 | 3.767 | 3.794 | 3.021 | 4.488 | 4.140 | 0.927 |
| 8.0 | 0.563 | 0.437 | 3.770 | 3.795 | 3.029 | 4.488 | 4.140 | 0.927 |
| 9.0 | 0.563 | 0.437 | 3.767 | 3.792 | 3.019 | 4.488 | **4.116** | 0.927 |
| 10.0 | 0.563 | 0.437 | 3.776 | 3.794 | 3.047 | 4.488 | 4.164 | 0.927 |

# E   Minor analysis of evaluation results

Interestingly, the Matcha-TTS (SFM) model trained on VCTK achieves a much lower WER than the ground-truth and vocoder-reconstructed speech. Since VCTK is a reading-style corpus, we observe that speakers occasionally exhibit disfluencies such as stammering or slurring. The synthesized speech, being generated from clean transcripts, avoids such inconsistencies.

For models trained on LibriTTS, vocoder-reconstructed speech performs significantly worse in SMOS tests. Since we adopt cross-sentence evaluations [8, 7], the prompt and target utterances are from the same speaker but are not necessarily adjacent, often resulting in prosodic mismatches. In contrast, CosyVoice LLM can imitate the prompt's prosody, and the generated speech tokens align well with the prompt. Ground truth still performs competitively, likely due to its superior speech quality.

Reconstructed speech also performs poorly in CMOS tests. As LibriTTS is from audiobooks, which often feature expressive and emotional delivery. We observe that listeners tend to perceive such utterances as unnatural when heard in isolation. While flow matching-based models are capable of generating high-quality speech, prosody becomes the dominant factor in listener ratings. Ground-truth speech can outperform synthesized speech in CMOS due to its better sound quality.

# F   RTF and NFE with using adaptive-step ODE solvers

In this section, we provide not only the RTFs for ODE solver inference in Table 10, but also the RTFs for the full model inference in Table 11. These comparisons reveal that the ODE solver inference dominates the overall inference time. When using adaptive-step ODE solvers, the inference acceleration of our SFM methods is significant.

Table 10: RTF and NFE results for adaptive-step ODE solvers (solver inference). $\overline{\text{RTF}}$ denotes the mean of RTF. Rate (%) denotes the relative speedup in terms of $\overline{\text{RTF}}$ compared to the ablated model.

| System | Heun(2) | | | Fehlberg(2) | | | Bosh(3) | | | Dopri(5) | | |
|---|---|---|---|---|---|---|---|---|---|---|---|---|
| | $\overline{\text{RTF}}\downarrow$ | Rate↑ | NFE↓ | $\overline{\text{RTF}}\downarrow$ | Rate↑ | NFE↓ | $\overline{\text{RTF}}\downarrow$ | Rate↑ | NFE↓ | $\overline{\text{RTF}}\downarrow$ | Rate↑ | NFE↓ |
| *Matcha-TTS trained on LJ Speech* | | | | | | | | | | | | |
| Baseline | 0.407 | -1.496 | 312.36 | 0.054 | 3.571 | 44.82 | 0.271 | -0.370 | 223.95 | 0.142 | 2.069 | 120.10 |
| Ablated | 0.401 | 0.000 | 306.72 | 0.056 | 0.000 | 45.28 | 0.270 | 0.000 | 221.81 | 0.145 | 0.000 | 121.46 |
| SFM ($\alpha$=1.0) | 0.361 | 9.858 | 277.56 | 0.053 | 4.171 | 44.08 | 0.225 | 16.924 | 186.23 | 0.132 | 9.100 | 111.14 |
| SFM ($\alpha$=2.0) | 0.299 | 25.541 | 229.43 | 0.048 | 13.306 | 39.16 | 0.187 | 30.931 | 153.08 | 0.115 | 20.484 | 96.02 |
| SFM ($\alpha$=3.0) | 0.249 | 37.931 | 191.49 | 0.043 | 22.749 | 34.88 | 0.157 | 41.895 | 129.02 | 0.100 | 30.753 | 84.20 |
| SFM ($\alpha$=4.0) | 0.207 | 48.486 | 156.78 | 0.037 | 34.246 | 29.74 | 0.131 | 51.576 | 107.90 | 0.086 | 40.559 | 72.56 |
| SFM ($\alpha$=5.0) | 0.157 | 60.825 | 120.07 | 0.027 | 50.968 | 22.14 | 0.110 | 59.266 | 90.74 | 0.076 | 47.605 | 63.74 |
| *Matcha-TTS trained on VCTK* | | | | | | | | | | | | |
| Baseline | 0.972 | -2.101 | 318.42 | 0.139 | -6.107 | 49.00 | 0.681 | 1.304 | 243.40 | 0.365 | 1.617 | 133.92 |
| Ablated | 0.952 | 0.000 | 313.18 | 0.131 | 0.000 | 45.52 | 0.690 | 0.000 | 245.69 | 0.371 | 0.000 | 134.76 |
| SFM ($\alpha$=1.0) | 0.950 | 0.210 | 312.26 | 0.123 | 6.269 | 42.60 | 0.690 | 0.030 | 244.43 | 0.371 | 0.111 | 134.00 |
| SFM ($\alpha$=2.0) | 0.771 | 18.998 | 253.66 | 0.111 | 15.463 | 38.28 | 0.558 | 19.071 | 196.58 | 0.318 | 14.356 | 114.08 |
| SFM ($\alpha$=3.0) | 0.626 | 34.271 | 205.49 | 0.100 | 23.635 | 34.50 | 0.436 | 36.844 | 153.86 | 0.265 | 28.626 | 95.48 |
| SFM ($\alpha$=4.0) | 0.519 | 45.449 | 170.63 | 0.093 | 29.503 | 31.96 | 0.350 | 49.257 | 123.80 | 0.231 | 37.846 | 83.48 |
| SFM ($\alpha$=5.0) | 0.407 | 57.242 | 133.78 | 0.073 | 44.413 | 25.10 | 0.279 | 59.547 | 99.05 | 0.195 | 47.454 | 70.28 |
| *StableTTS trained on VCTK* | | | | | | | | | | | | |
| Ablated | 2.153 | 0.000 | 712.06 | 0.394 | 0.000 | 133.96 | 1.270 | 0.000 | 434.74 | 0.998 | 0.000 | 346.12 |
| SFM ($\alpha$=1.0) | 1.212 | 43.694 | 397.26 | 0.261 | 33.707 | 88.36 | 0.790 | 37.823 | 269.32 | 0.690 | 30.849 | 232.60 |
| SFM ($\alpha$=2.0) | 0.982 | 54.386 | 316.00 | 0.232 | 41.301 | 76.48 | 0.697 | 45.123 | 231.82 | 0.613 | 38.593 | 208.72 |
| SFM ($\alpha$=3.0) | 0.789 | 63.346 | 254.74 | 0.205 | 48.081 | 67.44 | 0.611 | 51.930 | 204.64 | 0.564 | 43.473 | 190.60 |
| SFM ($\alpha$=4.0) | 0.592 | 72.497 | 192.34 | 0.169 | 57.149 | 53.84 | 0.542 | 57.305 | 181.72 | 0.489 | 51.003 | 166.12 |
| SFM ($\alpha$=5.0) | 0.586 | 72.792 | 189.30 | 0.173 | 56.096 | 53.64 | 0.538 | 57.674 | 178.90 | 0.505 | 49.377 | 169.48 |
| *CosyVoice trained on LibriTTS* | | | | | | | | | | | | |
| Baseline | 2.416 | 4.581 | 390.06 | 0.358 | 2.981 | 58.62 | 1.894 | 12.920 | 327.25 | 1.294 | 7.373 | 215.06 |
| Ablated | 2.532 | 0.000 | 395.25 | 0.369 | 0.000 | 58.91 | 2.175 | 0.000 | 346.90 | 1.397 | 0.000 | 223.37 |
| SFM ($\alpha$=1.0) | 1.745 | 31.093 | 275.92 | 0.283 | 23.449 | 45.68 | 1.454 | 33.143 | 232.67 | 1.061 | 24.069 | 170.57 |
| SFM ($\alpha$=2.0) | 1.426 | 43.667 | 218.17 | 0.250 | 32.196 | 39.63 | 1.211 | 44.326 | 192.97 | 0.917 | 34.395 | 146.30 |
| SFM ($\alpha$=3.0) | 1.110 | 56.182 | 173.28 | 0.219 | 40.682 | 34.85 | 1.001 | 53.952 | 158.57 | 0.787 | 43.679 | 126.53 |
| SFM ($\alpha$=4.0) | 0.963 | 61.967 | 149.95 | 0.186 | 49.642 | 29.98 | 0.887 | 59.217 | 142.15 | 0.720 | 48.469 | 116.09 |
| SFM ($\alpha$=5.0) | 0.960 | 62.096 | 149.90 | 0.185 | 49.875 | 29.96 | 0.888 | 59.190 | 141.79 | 0.727 | 47.990 | 116.15 |
| *CosyVoice-DiT trained on LibriTTS* | | | | | | | | | | | | |
| Ablated | 1.170 | 0.000 | 834.46 | 0.188 | 0.000 | 137.56 | 0.699 | 0.000 | 514.30 | 0.539 | 0.000 | 403.36 |
| SFM ($\alpha$=1.0) | 0.621 | 46.934 | 421.90 | 0.125 | 33.704 | 86.84 | 0.396 | 43.324 | 277.72 | 0.337 | 37.591 | 234.70 |
| SFM ($\alpha$=2.0) | 0.503 | 56.980 | 335.96 | 0.108 | 42.475 | 75.28 | 0.336 | 51.894 | 237.88 | 0.286 | 46.910 | 207.94 |
| SFM ($\alpha$=3.0) | 0.391 | 66.558 | 267.38 | 0.092 | 51.292 | 63.82 | 0.295 | 57.811 | 207.58 | 0.257 | 52.252 | 184.24 |
| SFM ($\alpha$=4.0) | 0.300 | 74.320 | 205.89 | 0.073 | 61.121 | 51.72 | 0.259 | 62.958 | 182.17 | 0.229 | 57.595 | 163.48 |
| SFM ($\alpha$=5.0) | 0.302 | 74.212 | 205.67 | 0.074 | 60.825 | 51.74 | 0.258 | 63.061 | 182.23 | 0.229 | 57.523 | 163.06 |

Table 11: RTF results when using adaptive-step ODE solvers (model inference). $\overline{\text{RTF}}$ denotes the mean of RTF. Rate (%) denotes the relative speedup compared to the ablated model.

| System | Heun(2) | | Fehlberg(2) | | Bosh(3) | | Dopri(5) | |
|---|---|---|---|---|---|---|---|---|
| | $\overline{\text{RTF}}\downarrow$ | Rate↑ | $\overline{\text{RTF}}\downarrow$ | Rate↑ | $\overline{\text{RTF}}\downarrow$ | Rate↑ | $\overline{\text{RTF}}\downarrow$ | Rate↑ |
| ***Matcha-TTS trained on LJ Speech*** | | | | | | | | |
| Baseline | 0.409 | -1.489 | 0.055 | 3.509 | 0.273 | -0.368 | 0.143 | 2.055 |
| Ablated | 0.403 | 0.000 | 0.057 | 0.000 | 0.272 | 0.000 | 0.146 | 0.000 |
| SFM ($\alpha$=1.0) | 0.363 | 9.809 | 0.055 | 4.017 | 0.226 | 16.814 | 0.133 | 8.976 |
| SFM ($\alpha$=2.0) | 0.300 | 25.430 | 0.050 | 12.866 | 0.188 | 30.735 | 0.117 | 20.230 |
| SFM ($\alpha$=3.0) | 0.251 | 37.773 | 0.045 | 22.063 | 0.159 | 41.639 | 0.102 | 30.399 |
| SFM ($\alpha$=4.0) | 0.208 | 48.282 | 0.038 | 33.261 | 0.133 | 51.268 | 0.088 | 40.104 |
| SFM ($\alpha$=5.0) | 0.159 | 60.578 | 0.029 | 49.541 | 0.112 | 58.914 | 0.077 | 47.076 |
| ***Matcha-TTS trained on VCTK*** | | | | | | | | |
| Baseline | 0.975 | -2.094 | 0.142 | -5.185 | 0.684 | 1.441 | 0.369 | 1.600 |
| Ablated | 0.955 | 0.000 | 0.135 | 0.000 | 0.694 | 0.000 | 0.375 | 0.000 |
| SFM ($\alpha$=1.0) | 0.953 | 0.211 | 0.127 | 6.111 | 0.693 | 0.036 | 0.374 | 0.116 |
| SFM ($\alpha$=2.0) | 0.775 | 18.925 | 0.115 | 15.046 | 0.562 | 18.971 | 0.321 | 14.213 |
| SFM ($\alpha$=3.0) | 0.629 | 34.141 | 0.104 | 23.000 | 0.439 | 36.651 | 0.268 | 28.349 |
| SFM ($\alpha$=4.0) | 0.523 | 45.274 | 0.096 | 28.711 | 0.354 | 48.998 | 0.234 | 37.481 |
| SFM ($\alpha$=5.0) | 0.411 | 57.021 | 0.077 | 43.209 | 0.283 | 59.235 | 0.199 | 46.993 |
| ***StableTTS trained on VCTK*** | | | | | | | | |
| Ablated | 2.157 | 0.000 | 0.398 | 0.000 | 1.274 | 0.000 | 1.001 | 0.000 |
| SFM ($\alpha$=1.0) | 1.216 | 43.622 | 0.265 | 33.396 | 0.793 | 37.716 | 0.694 | 30.730 |
| SFM ($\alpha$=2.0) | 0.986 | 54.294 | 0.235 | 40.912 | 0.701 | 44.992 | 0.616 | 38.447 |
| SFM ($\alpha$=3.0) | 0.793 | 63.242 | 0.208 | 47.643 | 0.614 | 51.781 | 0.568 | 43.315 |
| SFM ($\alpha$=4.0) | 0.596 | 72.380 | 0.172 | 56.640 | 0.546 | 57.146 | 0.492 | 50.821 |
| SFM ($\alpha$=5.0) | 0.589 | 72.674 | 0.177 | 55.598 | 0.541 | 57.513 | 0.509 | 49.200 |
| ***CosyVoice trained on LibriTTS*** | | | | | | | | |
| Baseline | 2.417 | 4.580 | 0.359 | 2.973 | 1.895 | 12.914 | 1.295 | 7.368 |
| Ablated | 2.533 | 0.000 | 0.370 | 0.000 | 2.176 | 0.000 | 1.398 | 0.000 |
| SFM ($\alpha$=1.0) | 1.746 | 31.076 | 0.284 | 23.353 | 1.455 | 33.124 | 1.062 | 24.035 |
| SFM ($\alpha$=2.0) | 1.428 | 43.644 | 0.252 | 32.070 | 1.212 | 44.301 | 0.918 | 34.360 |
| SFM ($\alpha$=3.0) | 1.111 | 56.155 | 0.220 | 40.536 | 1.003 | 53.922 | 0.788 | 43.638 |
| SFM ($\alpha$=4.0) | 0.964 | 61.937 | 0.187 | 49.471 | 0.888 | 59.184 | 0.721 | 48.424 |
| SFM ($\alpha$=5.0) | 0.961 | 62.066 | 0.186 | 49.705 | 0.889 | 59.157 | 0.728 | 47.946 |
| ***CosyVoice-DiT trained on LibriTTS*** | | | | | | | | |
| Ablated | 1.171 | 0.000 | 0.190 | 0.000 | 0.700 | 0.000 | 0.541 | 0.000 |
| SFM ($\alpha$=1.0) | 0.622 | 46.870 | 0.126 | 33.406 | 0.397 | 43.221 | 0.338 | 37.472 |
| SFM ($\alpha$=2.0) | 0.505 | 56.903 | 0.110 | 42.113 | 0.338 | 51.779 | 0.288 | 46.775 |
| SFM ($\alpha$=3.0) | 0.393 | 66.472 | 0.093 | 50.872 | 0.296 | 57.683 | 0.259 | 52.102 |
| SFM ($\alpha$=4.0) | 0.302 | 74.225 | 0.075 | 60.633 | 0.260 | 62.820 | 0.230 | 57.432 |
| SFM ($\alpha$=5.0) | 0.303 | 74.116 | 0.075 | 60.336 | 0.260 | 62.924 | 0.230 | 57.360 |

