# OpenReview forum: "Shallow Flow Matching for Coarse-to-Fine Text-to-Speech Synthesis"
_NeurIPS.cc/2025/Conference — NeurIPS 2025 poster_

### Official Review · Reviewer_Dwas · 2025-06-23

**Clarity:** 4
**Significance:** 3
**Originality:** 4
**Rating:** 5
**Confidence:** 4

**Summary:**

In this work, the authors present a novel Shallow Flow Matching (SFM) mechanism inspired by DiffSinger's [27] Shallow Diffusion Mechanism. SFM constructs intermediate paths along the Flow Matching's (FM) path, using coarse output representations i.e, it introduces a way of starting the ODE solver from the intermediate state in the flow path instead of Gaussian noise $N(0, I)$ as done by previous SOTA methods. Using the principle of shallow diffusion mechanism, they introduce an adaptive timestep $t_h$ which can be used as a pre-informed point to initiate solving the vector field ODE. During training, authors propose to use an orthogonal projection (by adding an SFM head to the encoder) method to predict the temporal parameter and use it to estimate the segment's starting point (coarse output). During inference, this coarse output is used as the initial point instead of isotropic Gaussian noise. Evaluations confirm that SFM improves both subjective, and objective evaluations and accelerates inference when using an **adaptive ODE solver**.

**Questions:**

* What are $\hat{t_g}$ and $\hat{\sigma_g}$ in Table 2? it is not concretely defined in the paper; are they already scaled?
* Practically, most of these architectures (at least all three of Matcha-TTS, Stable-TTS, and CosyVoice {which uses Matcha-TTS's U-Net}), in the architecture of their flow prediction network, concatenate the noise and coarse spectrogram ($\mu$). This essentially means the initial point is sampling from Gaussian noise centered around  $N(\mu, I)$. How is this different from such modifications? Is it that SFM also predicts a $\sigma$ so the noise is scaled at the coarse spectrogram?

**Ethical Concerns:**

["NO or VERY MINOR ethics concerns only"]

**Final Justification:**

The authors addressed my concerns; while the paper is grounded well theoretically, it holds less importance in practicality.

**Limitations:**

Yes

**Paper Formatting Concerns:**

Please define what are $\hat{t_g}$ and $\hat{\sigma_g}$ in Table 2 maybe in table caption?

**Quality:**

4

**Strengths And Weaknesses:**

Strengths
- Evaluating the proposed method on three different SOTA architectures, mainly consisting of UNets, and DiT, demonstrates the effectiveness of SFM and decouples the advantage of proposed changes with previous design choices.
- Novel and principled design improvement, by formalising piecewise-single segment flow and then evaluating it in practice.This grounds the advantage of using SFM especially when using adaptive ODE solver, showing that model can take longer steps confidently when solving the ODE as error is smaller than using just OT-CFM.

### Weakness
* While in theory method improves the solution of ODE when using an adaptive solver. In practice none of the SOTA models use an adaptive solver. Most use either order 1 or order 2 solvers and perceptually it makes little to no difference.
* Because of an additional hyperparameter $\alpha$ which is tuned for each architecture each dataset, the method adds an additional overhead to optimise this parameter.
* It would have been better if the authors could quantify Number of Function Evaluation (NFE), a commonly used evaluation term in diffusion/flow matching literature.

---

> ### Author Rebuttal · Authors · 2025-07-30
>
> We appreciate the reviewer's time and valuable feedback. It is our great honor to have the work carefully reviewed and considered. We provide our responses below.
>
> > **Q1:** What are $\hat{t_\boldsymbol{g}}$ and $\hat{\sigma_\boldsymbol{g}}$ in Table 2? it is not concretely defined in the paper; are they already scaled?
>
> **A1:** Thank you for pointing this out. This was a notational inconsistency in our submission. The $\hat{t_\boldsymbol{g}}$ and $\hat{\sigma_\boldsymbol{g}}$ in Table 2 correspond to $\tilde{t_{\boldsymbol{g}}}$ and $\tilde{\sigma_{\boldsymbol{g}}}$ defined in Equation (19). We will correct this notation and clarify their definitions in the final version to avoid confusion. And yes, they are already scaled.
>
> > **Q2:** Practically, most of these architectures (at least all three of Matcha-TTS, Stable-TTS, and CosyVoice {which uses Matcha-TTS's U-Net}), in the architecture of their flow prediction network, concatenate the noise and coarse spectrogram ($\mu$). This essentially means the initial point is sampling from Gaussian noise centered around $N(\mu, I)$. How is this different from such modifications? Is it that SFM also predicts a $\sigma$ so the noise is scaled at the coarse spectrogram?
>
> **A2:** Based on our experimental observations to date, we believe an answer to this question is that directly using the coarse output as an FM condition is less effective than using it to construct intermediate states. When constructing the intermediate states in the SFM method, we found that achieving better generation quality requires:
> 1. applying a nonlinear transformation to the coarse output;
> 2. strictly following the CondOT paths, as established by Theorems 1 and 2.
>
> Besides, as discussed on SFM-c in Section 5.1, using coarse output as an FM condition in SFM models either disrupts the SFM process during training or leads to degraded generation quality.
>
> Therefore, we hypothesize that directly conditioning on the coarse output may propagate its inherent errors and negatively affect the final output. In contrast, using it to construct intermediate states mitigates this issue by more effectively incorporating it into the FM process. A thorough validation of this hypothesis requires more dedicated experimental design and is beyond the scope of this paper.
>
> > **Q3 (Weakness 3):** It would have been better if the authors could quantify Number of Function Evaluation (NFE), a commonly used evaluation term in diffusion/flow matching literature.
>
> (We fully agree with the first two concerns raised in the weaknesses. These points are current limitations of our method and will serve as important directions for future improvement.)
>
> **A3:** We sincerely appreciate your recommendation and find the NFE to be a valuable metric, which has not been widely adopted in TTS research. We report the average number of calls to the flow module as NFE, and present the corresponding results in the table below. These values directly align with those in Table 10, which we omit here for clarity.
>
> We observe that NFE is more stable than RTF across different settings. However, since the actual computation time of each flow module call varies depending on the input sequence length, reporting both NFE and RTF in the paper offers more complete information.
>
>
> ### Matcha-TTS trained on LJ Speech
>
> | System | Heun(2) NFE↓ | Fehlberg(2) NFE↓ | Bosh(3) NFE↓ | Dopri(5) NFE↓ |
> |----------|----------|----------|----------|----------|
> | Ablated | 306.72 | 45.28 | 221.81 | 121.46 |
> | SFM ($\alpha$=1.0) | 277.56 | 44.08 | 186.23 | 111.14 |
> | SFM ($\alpha$=2.0) | 229.43 | 39.16 | 153.08 | 96.02 |
> | SFM ($\alpha$=3.0) | 191.49 | 34.88 | 129.02 | 84.20 |
> | SFM ($\alpha$=4.0) | 156.78 | 29.74 | 107.90 | 72.56 |
> | SFM ($\alpha$=5.0) | 120.07 | 22.14 | 90.74 | 63.74 |
>
> ### Matcha-TTS trained on VCTK
>
> | System | Heun(2) NFE↓ | Fehlberg(2) NFE↓ | Bosh(3) NFE↓ | Dopri(5) NFE↓ |
> |----------|----------|----------|----------|----------|
> | Ablated | 313.18 | 45.52 | 245.69 | 134.76 |
> | SFM ($\alpha$=1.0) | 312.26 | 42.60 | 244.43 | 134.00 |
> | SFM ($\alpha$=2.0) | 253.66 | 38.28 | 196.58 | 114.08 |
> | SFM ($\alpha$=3.0) | 205.49 | 34.50 | 153.86 | 95.48 |
> | SFM ($\alpha$=4.0) | 170.63 | 31.96 | 123.80 | 83.48 |
> | SFM ($\alpha$=5.0) | 133.78 | 25.10 | 99.05 | 70.28 |
>
> ### StableTTS trained on VCTK
>
> | System | Heun(2) NFE↓ | Fehlberg(2) NFE↓ | Bosh(3) NFE↓ | Dopri(5) NFE↓ |
> |----------|----------|----------|----------|----------|
> | Ablated | 712.06 | 133.96 | 434.74 | 346.12 |
> | SFM ($\alpha$=1.0) | 397.26 | 88.36 | 269.32 | 232.6 |
> | SFM ($\alpha$=2.0) | 316.0 | 76.48 | 231.82 | 208.72 |
> | SFM ($\alpha$=3.0) | 254.74 | 67.44 | 204.64 | 190.6 |
> | SFM ($\alpha$=4.0) | 192.34 | 53.84 | 181.72 | 166.12 |
> | SFM ($\alpha$=5.0) | 189.3 | 53.64 | 178.9 | 169.48 |
>
> ### CosyVoice trained on LibriTTS
>
> | System | Heun(2) NFE↓ | Fehlberg(2) NFE↓ | Bosh(3) NFE↓ | Dopri(5) NFE↓ |
> |----------|----------|----------|----------|----------|
> | Ablated | 395.25 | 58.91 | 346.90 | 223.37 |
> | SFM ($\alpha$=1.0) | 275.92 | 45.68 | 232.67 | 170.57 |
> | SFM ($\alpha$=2.0) | 218.17 | 39.63 | 192.97 | 146.30 |
> | SFM ($\alpha$=3.0) | 173.28 | 34.85 | 158.57 | 126.53 |
> | SFM ($\alpha$=4.0) | 149.95 | 29.98 | 142.15 | 116.09 |
> | SFM ($\alpha$=5.0) | 149.90 | 29.96 | 141.79 | 116.15 |
>
> ### CosyVoice-DiT trained on LibriTTS
>
> | System | Heun(2) NFE↓ | Fehlberg(2) NFE↓ | Bosh(3) NFE↓ | Dopri(5) NFE↓ |
> |----------|----------|----------|----------|----------|
> | Ablated | 834.46 | 137.56 | 514.30 | 403.36 |
> | SFM ($\alpha$=1.0) | 421.90 | 86.84 | 277.72 | 234.70 |
> | SFM ($\alpha$=2.0) | 335.96 | 75.28 | 237.88 | 207.94 |
> | SFM ($\alpha$=3.0) | 267.38 | 63.82 | 207.58 | 184.24 |
> | SFM ($\alpha$=4.0) | 205.89 | 51.72 | 182.17 | 163.48 |
> | SFM ($\alpha$=5.0) | 205.67 | 51.74 | 182.23 | 163.06 |

---

> > ### Comment · Reviewer_Dwas · 2025-08-05
> > **Thank you for additional results**
> >
> > Thank you for the further explanation and experiments. My comments have been addressed.

---

### Official Review · Reviewer_esVr · 2025-06-26

**Clarity:** 4
**Significance:** 3
**Originality:** 4
**Rating:** 5
**Confidence:** 4

**Summary:**

The authors propose a Shallow Flow Matching (SFM) algorithm that can inference from the intermediate state rather than pure noise and reduce computation cost and time for the Flow matching generation.  An orthogonal projection method to adaptively determine the temporal position of the intermediate states, and a single-segment piecewise flow is proposed to generate remaining states. Experiments show that SFM consistently improves the naturalness of synthesized speech in both objective and subjective evaluations. It is a solid work.

**Questions:**

1 In Table 3, the proposed systems are outperform than the baseline systems, which is derived from pure noise. Can we state the generation outputs are better if they were generated the intermediate states instead of noise states? Any explanation?

2. Table 4 and 10 compare the RTFs for different ODE solvers under different $\alpha$. They only compared with ablated systems, how about the comparison with the baselines mentioned?

**Ethical Concerns:**

["NO or VERY MINOR ethics concerns only"]

**Quality:**

3

**Strengths And Weaknesses:**

Strengths:
- An improved flow matching algorithm is proposed to reduce the computation.
- A proof is provided for the proposed algorithm
- Good results are achieved compared with multiple baselines

Weaknesses:
- The computation cost baseline only include the ablated systems it would be helpful if the baselines results are included too.

---

> ### Author Rebuttal · Authors · 2025-07-30
>
> We appreciate the reviewer's insightful and constructive comments. We provide our responses below.
>
> > **Q1:** In Table 3, the proposed systems are outperform than the baseline systems, which is derived from pure noise. Can we state the generation outputs are better if they were generated the intermediate states instead of noise states? Any explanation?
>
> **A1:** Based on our experiments to date, we believe this statement is valid. Generation from pure noise, as mentioned in our paper, suffers from "a suboptimal allocation of modeling capacity". However, in our preliminary experiments, we observed that the following naive approaches failed to improve and even degraded generation quality:
> 1. Simply adding noise to the coarse output as the initial or intermediate states;
> 2. Applying linear scaling to the coarse output and adding noise to obtain intermediate states.
>
> These results led us to realize that constructing effective intermediate states requires a nonlinear transformation of the coarse output, and must strictly follow the CondOT paths, as established by Theorems 1 and 2 in our paper. We hypothesize that this is due to the inherent mismatch between the training objectives of the coarse output and the flow matching process. Further improvements may be achievable by better balancing the associated losses during training, which we leave for future work.
>
> > **Q2:** Table 4 and 10 compare the RTFs for different ODE solvers under different $\alpha$. They only compared with ablated systems, how about the comparison with the baselines mentioned?
>
> **A2:** Thank you for your suggestion, which indeed improves the completeness of the tables. We have added the results of the three baseline models to Table 10, as shown below. The same as the original Table 10, "RTF" denotes the average real-time factor, and "Rate" (\%) indicates the relative speedup compared to the ablated model. We omit the std for clarity, as all std values are below 0.013.
>
> From the results, we observe the following:
> 1. The RTF differences between the baseline and ablated models are generally small.
> 2. For the CosyVoice trained on LibriTTS setting, the baseline model consistently shows slightly lower RTF values than the ablated model, suggesting that our modifications to this ablated model (also to the SFM model) slightly degrade the smoothness of the outputs during inference. To minimize architectural modifications, we adopted the simplest feature fusion strategy, as discussed in the CosyVoice part of Section 4.2. More sophisticated alternatives could potentially yield further improvements.
>
> ### Matcha-TTS trained on LJ Speech
>
> | System | Heun(2) RTF↓ | Heun(2) Rate↑ | Fehlberg(2) RTF↓ | Fehlberg(2) Rate↑ | Bosh(3) RTF↓ | Bosh(3) Rate↑ | Dopri(5) RTF↓ | Dopri(5) Rate↑ |
> |----------|----------|----------|----------|----------|----------|----------|----------|----------|
> | Baseline | 0.407 | -1.496 | 0.054 | 3.571 | 0.271 | -0.370 | 0.142 | 2.069 |
> | Ablated  | 0.401 | 0.000 | 0.056 | 0.000 | 0.270 | 0.000 | 0.145 | 0.000 |
> | SFM ($\alpha$=1.0) |        0.361 |         9.858 |            0.053 |               4.171 |      0.225 |        16.924 |         0.132 |           9.100   |
> | SFM ($\alpha$=2.0) |        0.299 |        25.541 |            0.048 |              13.306 |      0.187 |        30.931 |         0.115 |          20.484 |
> | SFM ($\alpha$=3.0) |        0.249 |        37.931 |            0.043 |              22.749 |      0.157 |        41.895 |         0.100 |          30.753 |
> | SFM ($\alpha$=4.0) |        0.207 |        48.486 |            0.037 |              34.246 |      0.131 |        51.576 |         0.086 |          40.559 |
> | SFM ($\alpha$=5.0) |        0.157 |        60.825 |            0.027 |              50.968 |      0.110 |        59.266 |         0.076 |          47.605 |
>
> ### Matcha-TTS trained on VCTK
>
> | System | Heun(2) RTF↓ | Heun(2) Rate↑ | Fehlberg(2) RTF↓ | Fehlberg(2) Rate↑ | Bosh(3) RTF↓ | Bosh(3) Rate↑ | Dopri(5) RTF↓ | Dopri(5) Rate↑ |
> |----------|----------|----------|----------|----------|----------|----------|----------|----------|
> | Baseline | 0.972 | -2.101 | 0.139 | -6.107 | 0.681 | 1.304 | 0.365 | 1.617 |
> | Ablated | 0.952 | 0.000 | 0.131 | 0.000 | 0.690 | 0.000 | 0.371 | 0.000 |
> | SFM ($\alpha$=1.0) | 0.950        | 0.210         | 0.123            | 6.269             | 0.690        | 0.030         | 0.371          | 0.111           |
> | SFM ($\alpha$=2.0) | 0.771        | 18.998        | 0.111            | 15.463            | 0.558        | 19.071        | 0.318          | 14.356          |
> | SFM ($\alpha$=3.0) | 0.626        | 34.271        | 0.100            | 23.635            | 0.436        | 36.844        | 0.265          | 28.626          |
> | SFM ($\alpha$=4.0) | 0.519        | 45.449        | 0.093            | 29.503            | 0.350        | 49.257        | 0.231          | 37.846          |
> | SFM ($\alpha$=5.0) | 0.407        | 57.242        | 0.073            | 44.413            | 0.279        | 59.547        | 0.195          | 47.454          |
>
> ### CosyVoice trained on LibriTTS
>
> | System | Heun(2) RTF↓ | Heun(2) Rate↑ | Fehlberg(2) RTF↓ | Fehlberg(2) Rate↑ | Bosh(3) RTF↓ | Bosh(3) Rate↑ | Dopri(5) RTF↓ | Dopri(5) Rate↑ |
> |----------|----------|----------|----------|----------|----------|----------|----------|----------|
> | Baseline | 2.416 | 4.581 | 0.358 | 2.981 | 1.894 | 12.920 | 1.294 | 7.373 |
> | Ablated | 2.532 | 0.000 | 0.369 | 0.000 | 2.175 | 0.000 | 1.397 | 0.000 |
> | SFM ($\alpha$=1.0) | 1.745        | 31.093        | 0.283            | 23.449            | 1.454        | 33.143        | 1.061          | 24.069          |
> | SFM ($\alpha$=2.0) | 1.426        | 43.667        | 0.250            | 32.196            | 1.211        | 44.326        | 0.917          | 34.395          |
> | SFM ($\alpha$=3.0) | 1.110        | 56.182        | 0.219            | 40.682            | 1.001        | 53.952        | 0.787          | 43.679          |
> | SFM ($\alpha$=4.0) | 0.963        | 61.967        | 0.186            | 49.642            | 0.887        | 59.217        | 0.720          | 48.469          |
> | SFM ($\alpha$=5.0) | 0.960        | 62.096        | 0.185            | 49.875            | 0.888        | 59.190        | 0.727          | 47.990          |

---

> ### Author Response · Authors · 2025-08-08
>
> Thank you again for your positive and constructive review. We truly appreciate the time and effort you have dedicated to assessing our work.
>
> If there is an opportunity to review our rebuttal, any additional thoughts would be greatly appreciated. Even brief feedback would help us ensure that our responses fully address your questions.

---

### Official Review · Reviewer_i9sS · 2025-07-03

**Clarity:** 2
**Significance:** 2
**Originality:** 2
**Rating:** 2
**Confidence:** 4

**Summary:**

This paper proposes Shallow Flow Matching (SFM), a mechanism designed to improve flow matching (FM)-based text-to-speech (TTS) models within a coarse-to-fine generation paradigm. SFM constructs intermediate states along the FM paths using coarse output representations. During training, it employs an orthogonal projection method to adaptively determine the temporal position of these states and applies a principled construction strategy based on a single-segment piecewise flow. During inference, SFM starts from this intermediate state instead of pure noise, focusing computation on the later stages of the FM paths. The authors integrate SFM into multiple TTS models.

**Questions:**

NA

**Ethical Concerns:**

["NO or VERY MINOR ethics concerns only"]

**Final Justification:**

My concerns regarding Q4 have been addressed, and Q1 (the novelty issue) has been partially addressed. However, I still believe that the other concerns (Q2.1, Q2.2, and Q3) remain unaddressed. Specifically:

1. **About the potential negative effect of the proposed method (Q3), and a lack of larger-scale experiments (Q2.1)**: Given the potential negative impact on WER, as demonstrated in the experimental results (Q3), a larger-scale experiment becomes especially crucial for validation (Q2.1).

2. **The proposed SFM could not be applied to single-stage FM models such as E2TTS and F5-TTS (Q2.2)**: I do not fully understand the reasons why E2-TTS and F5-TTS, two well-established flow-matching-based TTS models, cannot apply the SFM idea. By the way, I do not think the DiT or U-Net is the defining feature of these models (but the author claimed this during the response both to me and to the reviewer d41r). The true uniqueness lies in the single-stage (text to mel-spec directly) approach, rather than the two-stage (text-to-semantic + semantic-to-acoustic) approach. If by "coarse representations" the authors refer to the duration-aligned text representations (i.e., the frame-level linguistic representations), then E2-TTS and F5-TTS can also be seen as frame-level linguistic representations, albeit with many padding frames concatenated.

**Limitations:**

Yes

**Quality:**

2

**Strengths And Weaknesses:**

Strengths:
The motivation is clear. The problem that to speed up the FM-basd TTS models is worth to be explored.

Weaknesses:
1. As the authors mentioned it, the idea of shallow diffusion is proposed by DiffSinger. The unique challenge of introduce such idea into FM model has not be explained clearly.
2. The experimental settings is simple. Only three small-scale English dataset (LJSpeech, VCTK, and LibriTTS) are included. The authors could also introduce the single-stage FM model such as E2TTS or F5TTS as the base models.
3. Some results of Table 3 displayed that using SFM could lead to the negative effect for WER.
4. The presentation of the efficency benefits after using SFM is not clear. There should be also the RTF results in Table 3 in addition to Table 4.

---

> ### Author Rebuttal · Authors · 2025-07-30
>
> Thank you for your helpful comments. We address your concerns below:
>
> > **Q1:** As the authors mentioned it, the idea of shallow diffusion is proposed by DiffSinger. The unique challenge of introduce such idea into FM model has not be explained clearly.
>
> **A1:** While our SFM method is inspired by the idea of shallow diffusion (SD), the mechanisms are fundamentally different due to the differences between diffusion and flow matching (FM) processes:
> 1. **Timestep prediction:** Both SD and SFM need to find the timestep $t_\boldsymbol{h}$ at which the coarse output corresponds to the diffusion or FM trajectory. SD proposes a boundary predictor trained with a cross-entropy loss to classify whether a mel-spectrogram at any timestep $t$ comes from the coarse output or the diffusion process, and then identify the $t_\boldsymbol{h}$ (called $k$ in [27]) based on the most confident timesteps. In contrast, SFM directly projects the coarse output onto the conditional optimal transport (CondOT) paths using orthogonal projection, yielding $t_\boldsymbol{h}$ without classification tasks.
> 2. **Intermediate state construction:** In SD, once $t_\boldsymbol{h}$ is identified, the coarse output can be inserted into the diffusion process at $t_\boldsymbol{h}$ as the intermediate state. In SFM, however, we must consider the intrinsic noise $\sigma_\boldsymbol{h}$ of the coarse output and ensure that the constructed intermediate state lies strictly on the CondOT paths. To achieve this, we propose a principled formulation supported by Theorem 1.
> 3. **Single-segment training and inference:** SD can directly skip the initial diffusion steps during training and inference. In SFM, to enable this behavior while maintaining theoretical validity under FM, we need to derive a piecewise formulation as shown in Theorem 2.
> 4. **SFM strength:** Owing to the linearity of CondOT paths, SFM allows us to scale the coarse output during inference to strengthen its guidance, leading to improved generation quality. Whether a similar strategy can be applied to SD remains an open question.
>
> **In summary**, while the core idea of starting from a shallow timestep is shared, SFM is fundamentally constructed based on the CondOT paths. The challenges are unique to FM mechanisms and are addressed explicitly in our paper.
>
> > **Q2.1:** The experimental settings is simple. Only three small-scale English dataset (LJSpeech, VCTK, and LibriTTS) are included.
>
> **A2.1:** We adopt three widely used datasets that also serve as standard benchmarks in TTS research [12, 13, 15, 17-20, 27, 30, 33, 35, 39]: LJSpeech, VCTK, and LibriTTS, which span both single- and multi-speaker settings. LibriTTS, in particular, is a comparatively larger dataset with over 500 hours of speech. While we agree that super-large datasets would be valuable, we believe our current setup provides a firm and fair validation of the SFM method:
> 1. Our goal is to evaluate the generality and efficiency of SFM, not to propose a new zero-shot TTS system;
> 2. Conducting experiments on super-large datasets is beyond our available computational resources.
>
> > **Q2.2:** The authors could also introduce the single-stage FM model such as E2TTS or F5TTS as the base models.
>
> **A2.2:**
>
> 1. As indicated in the title and throughout the paper, our method is designed for the models that combine FM with a coarse-to-fine generation structure. This modeling paradigm is becoming increasingly popular, as it combines the strength of FM in high-quality generation with the contextual learning ability and functional versatility enabled by the coarse stage.
>
> 2. Single-stage FM models, such as E2-TTS or F5-TTS, do not produce coarse representations. Therefore, SFM cannot be applied to them, because SFM needs to use the coarse output to construct an intermediate state along the FM paths.
>
> 3. In our experimental settings, we have tried to utilize the limited open-source FM-based TTS models to broaden coverage in terms of input modalities and backbone architectures. We further adapted CosyVoice by replacing its FM module with DiT blocks (**F5-TTS has adopted**), resulting in the CosyVoice-DiT variant introduced in our paper.
> Therefore, within the coarse-to-fine FM-based paradigm, our method is general and applicable to multiple input types, backbone architectures (as mentioned by Reviewer Dwas), and coarse-to-fine paradigms, as summarized (NAR: non-autoregressive, AR: autoregressive):
>
> | Model name | Input | FM backbone | Coarse-to-fine paradigm |
> |------|------|------|------|
> | Matcha-TTS | Phoneme token | U-Net | NAR + FM |
> | StableTTS | Phoneme token | DiT | NAR + FM |
> | CosyVoice | Speech token | U-Net | AR + FM |
> | CosyVoice-DiT | Speech token | DiT | AR + FM |
>
> > **Q3:** Some results of Table 3 displayed that using SFM could lead to the negative effect for WER.
>
> **A3:** As we discussed in Section 5.1, the effect of SFM on WER is sometimes positive, sometimes negative. We believe this is because, in SFM, the coarse output is used to construct intermediate states on FM paths, rather than being used directly as a condition of FM. This shift may introduce some instability in word-speech alignment. This suggests that there is still room for improving alignment quality within the current framework.
>
> However, the improvements brought by SFM in both pseudo-MOS and subjective evaluations are consistent and substantial across all models, indicating strong perceptual benefits despite the observed WER variability.
>
> > **Q4:** The presentation of the efficency benefits after using SFM is not clear. There should be also the RTF results in Table 3 in addition to Table 4.
>
> **A4:** We appreciate the suggestion to include RTF results in Table 3. However, we would like to clarify the following points:
> 1. The acceleration effect of SFM only works when using adaptive-step ODE solvers. As shown in Table 1, we follow the original configurations of the models, some of which use fixed-step solvers (Euler). Therefore, including RTF values in Table 3 would result in many empty and meaningless entries.
> 2. Since acceleration is not the primary focus of this work and due to page limitations, Table 4 presents partial RTF results. As stated in its caption, **the complete results** are provided in Table 10 of Appendix F.
> 3. In Table 10, we detail the speed-up achieved by SFM under various adaptive-step solvers and show that the acceleration consistently increases with larger values of $\alpha$, which the table is specifically designed to highlight.
>
> We appreciate your comments and hope these clarifications help convey the motivation, novelty, and effectiveness of our work more clearly.

---

> ### Comment · Reviewer_i9sS · 2025-08-04
> **Response to the Rebuttals**
>
> I would like to thank the authors for their efforts and the clarifications provided. My concerns regarding Q4 have been addressed, and Q1 has been partially addressed. However, I still believe that the other concerns (Q2.1, Q2.2, and Q3) remain unaddressed. Specifically:
>
> - **Q2.1 and Q3**: Given the potential negative impact on WER, as demonstrated in the experimental results (Q3), a larger-scale experiment becomes especially crucial for validation (Q2.1).
>
> - **Q2.2**: First, I recommend that the authors avoid using the term "coarse representations" in the next version of the paper, as it is overly ambiguous and could lead to confusion, particularly in the context of diffusion-related TTS tasks. Additionally, I do not fully understand the reasons why E2-TTS and F5-TTS, two well-established flow-matching-based TTS models, cannot apply the SFM idea. By the way, I do not think the DiT or U-Net is the defining feature of these models. The true uniqueness lies in the single-stage (text to mel-spec directly) approach, rather than the two-stage (text-to-semantic + semantic-to-acoustic) approach. If by "coarse representations" the authors refer to the duration-aligned text representations (i.e., the frame-level linguistic representations), then E2-TTS and F5-TTS can also be seen as frame-level linguistic representations, albeit with many padding frames concatenated. If the proposed method truly cannot be applied to single-stage flow-matching-based TTS models, this would represent another weakness of this study.
>
> Given these reasons, I will maintain my original rating.

---

> ### Author Response · Authors · 2025-08-04
>
> We appreciate your feedback and the opportunity to clarify the remaining concerns.
>
> **A1:** We would appreciate it if you could clarify which part of Q1 remains unaddressed. In short, a direct and simple answer to this question is that the challenge lies in the fundamental difference between diffusion and FM. We have attempted to provide a thorough explanation and would be happy to elaborate further based on your specific concerns.
>
> **A2.2:** Thank you for your response. It seems we now understand what caused the confusion.
>
> 1. The term "coarse representations" is associated with coarse-to-fine generation frameworks, which is a growing research interest. Here, **coarse-to-fine is not diffusion or FM process, but a multi-stage process involving multiple generative modules**. In other words, there is a weak generator before the FM module, and we have defined $\boldsymbol{g_\omega}$ as the weak generator in Equation (10) of our paper. We did not anticipate that "coarse representations" could be misunderstood as being related to diffusion. Recent paper like [31] has already used the term "coarse mel-spectrograms", which we also used in the content related to TTS experiments. In our next version, we will improve it to avoid any confusion.
>
> 2. Similarly to shallow diffusion, the SFM method is designed for multi-stage architectures, where the FM module is used in the final stage. In our paper, the **coarse mel-spectrograms** (not frame-level linguistic representations, also similar to shallow diffusion) generated by the weak generator are used to construct the intermediate states along the flow paths, thereby improving the quality of FM generation. In other words, we use the **early-stage output (from the weak generator, not the FM module)** to build the intermediate state along the flow paths, and the FM inference can start from this state, but not from pure noise. In contrast, the single-stage models only have one module and can not be applied with the SFM method.
>
> 3. In fact, single-stage FM-based models such as E2-TTS and F5-TTS are not widely adopted in recent TTS research, because FM itself is not good at contextual learning. Not only the FM-based TTS models [1, 2, 8, 9, 13, 15, 30, 31] mentioned in our paper, but also the latest SOTA TTS models with FM modules, such as CosyVoice3 [a], MiniMax-Speech [b], and Step-Audio 2 [c], have adopted multi-stage frameworks. Moreover, multi-modal models with FM modules like Qwen2.5-Omni [d] also benefit from multi-stage architectures, further highlighting both the relevance and scalability of our proposed method.
>
> **Our concern:** It appears that there may still be some misunderstandings about our method. If you could point out which part of the algorithm remains unclear or difficult to follow, we would be very grateful to clarify further or modify it in our next version.
>
> [a] Z. Du et al., "CosyVoice 3: Towards In-the-wild Speech Generation via Scaling-up and Post-training", arXiv:2505.17589, 2025. \
> [b] B. Zhang et al., "MiniMax-Speech: Intrinsic Zero-Shot Text-to-Speech with a Learnable Speaker Encoder", arXiv:2505.07916, 2025. \
> [c] B. Wu et al., "Step-Audio 2 Technical Report", arXiv:2507.16632, 2025. \
> [d] J. Xu et al., "Qwen2.5-Omni Technical Report", arXiv:2503.20215, 2025.

---

> > ### Comment · Reviewer_i9sS · 2025-08-04
> > **Comments to the Response**
> >
> > The claim "FM itself is not good at contextual learning" is certainly outside my understanding. This is my final comment. I'll leave the ultimate decision to the Area Chair.

---

### Official Review · Reviewer_d41r · 2025-07-03

**Clarity:** 2
**Significance:** 1
**Originality:** 2
**Rating:** 3
**Confidence:** 2

**Summary:**

This paper proposes a Shallow Flow Matching (SFM) mechanism to enhance flow matching (FM)-based Text-to-Speech (TTS) models that operate within a coarse-to-fine generation paradigm. The core idea is to leverage the coarse representations produced by a weak generator to construct an intermediate starting point for the FM process, rather than starting from pure noise. The authors demonstrate that integrating SFM into several TTS models improves both the sampling quality and inference speed.

**Questions:**

The main issues I would like to see addressed are already listed in the weaknesses section above.

**Ethical Concerns:**

["NO or VERY MINOR ethics concerns only"]

**Final Justification:**

I have raised my score from 2 to 3 based on the authors' rebuttal.

* The authors' rebuttal clarified the technical novelty of their method compared to other acceleration techniques and justified their experimental scope.
    * large-scale experiments are not necessary, but it would certainly help assess the proposed method's applicability to recent TTS systems that can follow highly diverse speaker prompts if existed.
* The primary weakness is the work's limited applicability. Its contribution is confined to a specific class of TTS models.

**Limitations:**

yes

**Quality:**

2

**Strengths And Weaknesses:**

Strengths:
* The paper provides a theoretical validation for the proposed SFM method, with theorems and proofs.
* The method demonstrates improved performance compared to its baseline counterparts in subjective evaluations of naturalness and similarity.
* The significant improvement in inference speed when using the proposed method is thoroughly analyzed.

Weaknesses:
* The experimental results on small datasets like LJSpeech and VCTK do not adequately justify the method's scalability. While LibriTTS is a comparatively larger set, the experiments do not convincingly demonstrate how this approach would perform on the massive, diverse datasets (10k+ hours) used by many state-of-the-art zero-shot TTS models (e.g., VALL-E, Voicebox, MELLE, DiTTo-TTS, E2-TTS, F5-TTS, and CosyVoice 2).
* The method has limited applicability in its current form. While the authors show its applicability to CosyVoice, a two-stage coarse-to-fine generative model, it is unclear if or how SFM could be applied to other prominent TTS methods, such as single-stage large diffusion/flow-matching models (e.g., DiTTo-TTS, F5-TTS) or discrete-token generative models (e.g., Llasa, MaskGCT).
* The paper does not compare the proposed acceleration technique to other well-known methods for speeding up generative model inference, such as consistency models. This makes it difficult to assess the significance of the speed improvements in the broader context of fast sampling techniques.

[1] Wang, Yuancheng, et al. "MaskGCT: Zero-Shot Text-to-Speech with Masked Generative Codec Transformer." The Thirteenth International Conference on Learning Representations.

---

> ### Author Rebuttal · Authors · 2025-07-30
>
> Thank you for your valuable comments. We address your concerns below:
>
> > **Q1:** The experimental results on small datasets like LJSpeech and VCTK do not adequately justify the method's scalability. While LibriTTS is a comparatively larger set, the experiments do not convincingly demonstrate how this approach would perform on the massive, diverse datasets (10k+ hours) used by many state-of-the-art zero-shot TTS models (e.g., VALL-E, Voicebox, MELLE, DiTTo-TTS, E2-TTS, F5-TTS, and CosyVoice 2).
>
> **A1:** We aim to evaluate the general effectiveness of the proposed method on various models rather than develop a new SOTA zero-shot TTS model. To this end, we adopt three widely used datasets that also serve as standard benchmarks in TTS research [12, 13, 15, 17-20, 27, 30, 33, 35, 39]: LJSpeech, VCTK, and LibriTTS, which span both single- and multi-speaker settings. LibriTTS, as you mentioned, is a comparatively larger dataset.
>
> Conducting experiments on super massive-scale corpora (10k+ hours) is typically done when the goal is to develop an SOTA zero-shot TTS model, which not only deviates from the intended scope of our work but is also far beyond our available computational resources. We believe our current setup provides a strong and fair validation of the SFM method.
>
> > **Q2:** The method has limited applicability in its current form. While the authors show its applicability to CosyVoice, a two-stage coarse-to-fine generative model, it is unclear if or how SFM could be applied to other prominent TTS methods, such as single-stage large diffusion/flow-matching models (e.g., DiTTo-TTS, F5-TTS) or discrete-token generative models (e.g., Llasa, MaskGCT).
>
> **A2:**
> 1. As indicated in the title and throughout the paper, the SFM method is designed for models that combine flow matching (FM) with a coarse-to-fine generation structure. Therefore, our method is not designed for the following situations: \
>     &nbsp;&nbsp;&nbsp;&nbsp; a. single-stage architectures such as DiTTo-TTS and F5-TTS, which do not produce coarse representations; \
>     &nbsp;&nbsp;&nbsp;&nbsp; b. discrete-token generative models such as Llasa or MaskGCT, which operate entirely in discrete latent spaces and do not employ FM at all. \
> Single-stage models often lack strong contextual modeling and functional versatility, while discrete-token-only models tend to underperform in generation quality compared to those using continuous representations. Due to these limitations, many generative models have begun to use FM within coarse-to-fine multi-stage frameworks. Our proposed SFM method is motivated by this emerging trend, aiming to further enhance this generation paradigm.
> 2. As researchers in the TTS field, we develop the SFM method and validate it with TTS models. Since FM has just shown strong potential in TTS, the FM-based **open-source** TTS models are limited, among them: \
>     &nbsp;&nbsp;&nbsp;&nbsp; a. F5-TTS [7], as mentioned above, adopts DiT blocks as its backbone and is in a single-stage architecture; \
>     &nbsp;&nbsp;&nbsp;&nbsp; b. VoiceFlow [15] adopts coarse-to-fine generation but has almost the same architecture as Matcha-TTS [30]. \
> As of the submission deadline, no other publicly available FM-based TTS models were found. To broaden coverage in terms of input modalities and backbone architectures, we further adapted CosyVoice by replacing its FM module with DiT blocks (**F5-TTS has adopted**), resulting in the CosyVoice-DiT variant introduced in our paper.
> Therefore, within the coarse-to-fine FM-based paradigm, our method is general and applicable to multiple input types, backbone architectures (as mentioned by Reviewer Dwas), and coarse-to-fine paradigms, as summarized (NAR: non-autoregressive, AR: autoregressive):
>
> | Model name | Input | FM backbone | Coarse-to-fine paradigm |
> |------|------|------|------|
> | Matcha-TTS | Phoneme token | U-Net | NAR + FM |
> | StableTTS | Phoneme token | DiT | NAR + FM |
> | CosyVoice | Speech token | U-Net | AR + FM |
> | CosyVoice-DiT | Speech token | DiT | AR + FM |
>
> > **Q3:** The paper does not compare the proposed acceleration technique to other well-known methods for speeding up generative model inference, such as consistency models. This makes it difficult to assess the significance of the speed improvements in the broader context of fast sampling techniques.
>
> **A3:** The acceleration brought by SFM is **fundamentally different** from that of consistency models:
> 1. SFM leads to acceleration only when adaptive-step ODE solvers are used. The acceleration arises from skipping the early part of the FM path, which results in smoother inference samples and fewer sampling steps. It is important to note that the generation quality is improved.
> 2. Consistency models [a], and their recent evolutions, shortcut models [b] and MeanFlow [c], work by adding regularization terms to the training objective, implicitly performing self-distillation. These methods enable a few or only one sampling step during inference, but typically come at the cost of reduced generation quality.
>
> Therefore, the above acceleration algorithms and SFM are not mutually exclusive and cannot be compared with each other. In contrast, we can directly use them at the same time for further efficiency gains. Since SFM skips the early part of the FM paths, the latter part is the standard FM process and is fully compatible with other acceleration methods on FM. We will further clarify this relationship in the "Related Works" section.
>
> [a] Y. Song et al., "Consistency Models", ICML, 2023. \
> [b] K. Frans et al., "One Step Diffusion via Shortcut Models", ICLR, 2025. \
> [c] Z. Geng et al., "Mean Flows for One-step Generative Modeling", arXiv:2505.13447, 2025.
>
> We appreciate your feedback and hope these clarifications help position our method more clearly within the TTS literature.

---

> ### Author Response · Authors · 2025-08-07
>
> Thank you again for your comments. We would greatly appreciate it if you could share your thoughts on our response, as your feedback would be helpful in further clarifying any remaining concerns.
>
> We realize that we may not have sufficiently emphasized in the introduction that the term coarse-to-fine inherently refers to a multi-stage generation paradigm, where a weak generator first produces coarse mel-spectrograms that are subsequently refined by the flow matching module.
>
> If anything remains unclear or confusing, we would be more than happy to engage in further discussion and provide additional clarification.

---

> ### Comment · Reviewer_d41r · 2025-08-07
>
> I thank the authors for their detailed response, which has clarified several key aspects of their work.
>
> The rebuttal has successfully convinced me on two points. First, the explanation of how SFM differs from other acceleration techniques like consistency models is clear; I now understand that SFM improves quality while accelerating adaptive solvers, making it a distinct contribution. Second, although large-scale experiments would certainly help assess the proposed method's applicability to recent TTS systems that can follow highly diverse speaker prompts, I accept the authors' position that evaluating on massive datasets to achieve state-of-the-art zero-shot performance is not essential to prove the efficacy of their proposed mechanism for the coarse-to-fine TTS systems.
>
> However, my primary concern regarding the work's limited applicability (W2) remains. The authors' response confirms that SFM is intentionally designed only for a specific paradigm: multi-stage, coarse-to-fine, continuous-valued TTS. While the method is effective within this architecture, the TTS field is actively exploring diverse approaches, including single-stage or discrete-token-only models. By design, this work does not impact these other prominent research directions, which fundamentally limits its overall significance and potential influence.
>
> In summary, the authors have proposed a technically sound method that demonstrably improves efficiency and performance for a specific class of popular TTS approaches. While large-scale experiments would still be valuable to understand the method's relevance to modern zero-shot systems, the limited applicability is the most significant remaining limitation.
>
> Based on these clarifications, I am raising my score to 3.

---

### Note · Authors · 2025-08-15

We believe that Reviewer Dwas has provided an excellent summary of our proposed shallow flow matching (SFM) method. Both Reviewer Dwas and Reviewer esVr have correctly captured the essence of our method:

When a flow matching (FM) module serves as a refiner to a weak generator, transforming the coarse representation into an intermediate state along the FM paths, and starting training and inference from this state rather than pure noise, yields better results than directly using the coarse representation as a condition.

**Regarding the scope** (primary concern of Reviewer d41r), we would like to add:
1. Our study focuses on cases where the FM module serves as a refiner. As a TTS research team, we validated the method on TTS tasks. However, the proposed framework and theoretical foundation are general and can inspire other tasks involving FM refiners (as stated in Appendix G).

2. While coarse-to-fine paradigms are common in TTS, FM has only begun to attract enough attention in this field. Publicly available TTS models with an FM module remain limited, making this an emerging research direction.

**Regarding the dataset scale**:

Beyond our stated reasons that our goal is to validate the method rather than to pursue a new SOTA zero-shot TTS model, we have to note that CosyVoice, the zero-shot TTS model used in our experiments, adopts LibriTTS as its standard dataset in its official open-source release, with corresponding configurations. (Recently a 755-hour Chinese dataset was also added.) Moreover, the LibriTTS test set (or LibriSpeech, from which LibriTTS is derived) remains a standard English benchmark for zero-shot TTS models, making our choice a widely accepted practice.

**Regarding the absence of E2-TTS (not officially open-sourced) and F5-TTS experiments**:

Only Reviewer i9sS appeared to misunderstand this point now. These single-stage models lack weak generators, making our method inapplicable. While F5-TTS inherits E2-TTS’s text representation approach, its contribution to FM lies in adopting DiT blocks and specific sampling strategy. To address this, we have already replaced CosyVoice’s U-Net FM module with DiT blocks, introducing the CosyVoice-DiT variant to complete our experimental setting.

---

### Decision · Program_Chairs · 2025-09-17

**Decision:**

Accept (poster)

**Comment:**

This paper introduces Shallow Flow Matching (SFM), an extension of flow matching, where inference begins from intermediate states rather than pure noise. Evaluations across multiple TTS models show consistent gains in generation quality and efficiency, with additional experiments (e.g., NFE results) strengthening the empirical evidence. Two reviewers raised concerns about dataset scale and limited practical significance, but these do not undermine the core contribution. The authors argue that the datasets used are standard in TTS research. The authors have also explained the choice of baselines. These seem to be reasonable. Moreover, while FM is not yet the default in practice, showing the use of shallow FM here is interesting. Given the novelty, and improvements, I conclude that the strengths outweigh these concerns and recommend acceptance of the work.